# TDRM: Smooth Reward Models with Temporal Difference for LLM RL and Inference

## Abstract

Reward models are central to both reinforcement learning (RL) with language models and inference-time verification. However, existing reward models often lack temporal consistency, leading to ineffective policy updates and unstable RL training. We introduce TDRM, a method for learning smoother and more reliable reward models by minimizing temporal differences (TD) for training-time reinforcement learning and inference-time verification. Experiments show that TD-trained process reward models (PRMs) improve performance across Best-of-$N$ (up to 6.6%) and tree-search (up to 23.7%) settings. When combined with Reinforcement Learning with Verifiable Rewards (RLVR), TD-trained PRMs lead to more data-efficient RL — achieving comparable performance with just 2.5k data to what baseline methods require 50.1k data to attain — and yield higher-quality language model policies in 8 model variants (5 series), e.g., Qwen2.5-(0.5B, 1,5B), GLM4-9B-0414, GLM-Z1-9B-0414, Qwen2.5-Math-(1.5B, 7B), and DeepSeek-R1-Distill-Qwen-(1.5B, 7B). All code is available at `https://anonymous.4open.science/r/TDRM-CDD6`.

## 1 Introduction

Reward Models (RMs), which provide rewards for the intermediate/final reasoning processes of Large Language Models (LLMs) [1; 36; 8], have now become a standard practice for LLM reasoning in post-training [28; 41; 53], demonstrating remarkable performance in various fields, including mathematical problem-solving [52; 51], code synthesis [55; 43], and instruction following [3; 25]. In particular, in mathematical reasoning, extensive research has explored how RMs benefit from fine-grained supervision at intermediate reasoning steps, giving rise to Process Reward Models (PRMs) [23] that leverage such step-wise signals, as opposed to Outcome Reward Models (ORMs) [23] relying solely on final-answer correctness. Reward models offer important advantages: *(1) RMs provide low-cost feedback signals compared to expensive human annotations, (2) PRMs enable intermediate-stage reward beyond the typically sparse signals from human or rule-based verifiers*. Furthermore, during online Reinforcement Learning (RL) training [32], process or rule-based reward mechanisms are crucial in enhancing LLM performance by providing effective feedback that guides reasoning quality.

However, a key limitation of current RMs lies in their *lack of temporal consistency*: the reward assigned to a given step in the reasoning trajectory is often unrelated to the reward at adjacent steps. For example, existing works [23; 6] tend to assign a single scalar value to an entire reasoning trajectory via PRM or ORM, without distinguishing beneficial or suboptimal intermediate steps. Qwen2.5-Math-PRM [54] explicitly argues against using Monte Carlo (MC) estimation for PRMs, as it relies on final trajectory outcomes to evaluate intermediate steps, leading to inaccurate step verification (e.g., correct answers from incorrect steps or vice versa). Meanwhile, models like Generalist Reward Modeling (GRM) [25] often fail to update rewards for current steps by incorporating context from preceding or subsequent steps when generating multi-step reasoning. This makes it difficult for RMs to distinguish how much each reasoning step contributes to final success, resulting in inconsistent and misleading reward feedback that degrades both *training-time learning signal* (e.g., in RL) and *inference-time search efficiency* (e.g., by encouraging suboptimal trajectories). These challenges are particularly pronounced in long chain-of-thought (CoT) scenarios (e.g., o1 [29], R1 [5; 47]), where models receive no reward until completing a long sequence of reasoning steps.

To tackle these challenges, we introduce TDRM that employs Temporal Difference (TD) learning for reward modeling (Figure 1). Unlike the prior approach [7] where TD was used to construct

Figure 1: Overall framework of TDRM. In panel (a), we employ $n$-step TD learning for training the PRM. In panel (b), process reward and verifiable reward are effectively combined for RL training. In panel (c), RL training results in Table 4 compare baselines against TDRM. Best-of-1024 outcomes in Table 2 contrast ScalarORM with TDRM (3-step TD). For greedy search evaluations in Figure 3, Qwen2.5-Math-7B with a branch factor of 8 is used to compare ScalarORM against TDRM (TD(2)).

offline datasets, our method leverages TD for online training, dynamically bootstrapping intermediate rewards by integrating future estimates at each step to derive process reward models. Additionally, we propose a strategy that takes advantage of both rule-based rewards (e.g., from Group Relative Policy Optimization (GRPO) [32]) and process rewards generated by TDRM, delivering denser reward signals for online RL training. We evaluate TDRM in two scenarios: *(1) inference-time verification* and *(2) training-time reinforcement learning*. Experimental results show that TDRM induces *smoother* reward landscapes compared to conventional PRM training — increasing the low rewards and reducing the high rewards — thus significantly improving verification accuracy (e.g., Best-of-$N$, tree search) during inference. TDRM also demonstrates enhanced RL performance, outperforming multiple LLM baselines in both reward signal density and learning efficiency on mathematical benchmarks.

In summary, our key contributions are listed below:

- We introduce the framework TDRM, aiming to learn more reliable reward models in RL training. By leveraging temporal difference learning, TDRM generates smoother reward landscapes in Figure 8.

- Training-time RL experiments show that incorporating TDRM into the RL loop yields strong performance gains (up to 51.1%) and data efficiency (matching 50.1k baseline performance with only 2.5k data) on 8 model variants (5 series) with an effective combination of verifiable rule-based and process rewards in Table 4.

- Inference-time verification demonstrates that online TD-trained PRMs significantly enhance performance in both Best-of-$N$ (up to 6.6%) in Table 2 and tree-search (up to 23.7%) in Figure 3.

## 2 PRELIMINARIES

### 2.1 LLM REASONING AS MDP

The reasoning process in LLMs can be framed as a Markov Decision Process (MDP) [39]. An MDP typically involves a state space $\mathcal{S}$, containing the full set of possible situations, and an action space $\mathcal{A}$, encompassing the set of allowable decisions. It also includes a transition function $f\colon \mathcal{S} \times \mathcal{A} \to \mathcal{S}$, along with a reward function $R\colon \mathcal{S} \times \mathcal{A} \to \{r \mid r \in [0,1]\}$. In our context, the state space corresponds to every possible token sequence generated so far, whereas the action space comprises all possible tokens that can be selected next [34]. The transition function $f$ in our setting is simply the concatenation operation $f(s_t, a_t) = s_t \cdot a_t$, where $\cdot$ denotes concatenation. Regarding LLM reasoning, the input prompt is given as $(q_0, \cdots, q_L)$, and at step $t - 1$, the sequence of generation tokens for a single solution is $(o_0, \cdots, o_{t-1})$. Thus, given a prompt, action $a_t$ is a newly generated token and $s_t$ is the token sequence or the context for LLM, i.e., $s_t = (q_0, \cdots, q_L, o_0, \cdots, o_{t-1})$. In our work, an action is defined as a newly generated sentence. In standard RL, the reward function $R(s_t, a_t)$ is designed to assign an expected value for the partial generation paths based on each $(s_t, a_t)$ pair. In our study, we particularly emphasize establishing a process reward signal at each step

$t$ and outcome reward in the terminal step $T$ to guide the judgment (reflecting the correctness of a partial reasoning trace) of generation (guiding the learning direction) of LLM.

## 2.2 REWARD MODELING FOR LLMS

Recent studies [23; 52] model process rewards by training intermediate steps with labeled (Discriminative/Scalar) or generated (Generative) rewards and outcome rewards by comparing the final output with ground truth. Specifically, *Process Reward Modeling* estimates the rewards of intermediate steps as hard or soft values using learning a value function or training a value network. In contrast, *Rule-based/Outcome Reward Modeling* obtains the outcome reward using a rule-driven function that allocates rewards exclusively according to whether the complete sequence is correct. In domains such as mathematical reasoning, code generation, and theorem proving, leveraging the final accuracy of verifiable tasks as an outcome reward has proven effective in strengthening reasoning abilities. Specifically, a correct output will receive a $+1$ reward, while an incorrect output will receive a $0$ reward. The goal of reward modeling is to help *generalize* to unseen, out-of-distribution (OOD) problems and provide guidance in such OOD scenarios.

## 2.3 ONLINE RL TRAINING

In this work, we adopt the zero RL training strategy [50] described in DeepSeek-R1 [5]. This approach utilizes GRPO [32], which removes the need for explicit value and advantage functions [9]. GRPO uses group-normalized rewards to estimate the advantages to further optimize computational efficiency. For a given query $q$, and the responses $O = o_1, o_2, \ldots, o_G$ are produced by the previous policy model $\pi_{\text{old}}$. The objective of GRPO is to refine the policy model $\pi$ as follows:

$$
\mathcal{J}_{\text{GRPO}}(\theta) = \mathbb{E}_{(q,a)\sim\mathcal{D},\{o_i\}_{i=1}^G \sim \pi_{\theta_{\text{old}}}(\cdot|q)} \left[ \frac{1}{G} \sum_{i=1}^{G} \frac{1}{|o_i|} \sum_{j=1}^{|o_i|} \left( \min \left( \frac{\pi_\theta(o_{i,j}|q, o_{i,<j})}{\pi_{\theta_{\text{old}}}(o_{i,j}|q, o_{i,<j})} \hat{A}_{i,j}, \right. \right. \right.
$$
$$
\left. \left. \left. \text{clip}\left(\frac{\pi_\theta(o_{i,j}|q, o_{i,<j})}{\pi_{\theta_{\text{old}}}(o_{i,j}|q, o_{i,<j})}, 1-\varepsilon, 1+\varepsilon\right)\hat{A}_{i,j}\right) - \beta D_{\text{KL}}(\pi_\theta||\pi_{\text{ref}})\right)\right], \tag{1}
$$

where $\pi_{\text{ref}}$ is the reference model, $o_{i,j}$ represents the token produced at $j$-th generation step in the $i$-th generated response. To limit deviation from the reference, a KL-divergence regularization term, $D_{\text{KL}}$, is incorporated. The advantage estimate $\hat{A}_{i,j}$ quantitatively reflects how much each response $o_i$ surpasses the group average. This is achieved by normalizing the reward within the group: $\hat{A}_{i,j} = \frac{r_i - \text{mean}(\{r_1, r_2, \ldots, r_G\})}{\text{std}(\{r_1, r_2, \ldots, r_G\})}$. $\frac{\pi_\theta(o_{i,j}|q, o_{i,<j})}{\pi_{\theta_{\text{old}}}(o_{i,j}|q, o_{i,<j})}$ denotes the likelihood ratio.

## 3 THE TDRM METHOD

TDRM employs temporal difference learning to construct reliable reward models for RL training, and can be integrated with verifiable rewards. The framework comprises three components (Figure 1):

- PRM Module: A process reward model trained via *n*-step TD learning with reward shaping.

- RL Module: Online RL guided by the trained process reward model to optimize policy updates.

- TDRM Integration: An effective linear combination of process reward from PRM and verifiable reward, applied to actor-critic style online RL across different policy model series and sizes.

## 3.1 UNDERSTANDING REWARD SMOOTHNESS

**Background.** Temporal difference (TD) methods enable the iterative refinement of policy value estimates by leveraging the inter-dependencies between states. In particular, $n$-step TD updates extend this concept by incorporating rewards and value estimates from $n$ subsequent states, providing a more comprehensive and forward-looking perspective compared to traditional 1-step TD. This approach discounts future rewards exponentially using a factor (e.g., $\gamma$, usually less than 1) to encourage receiving earlier rewards and balance short-term gains with long-term consequences of actions.

In the context of LLM reasoning, each step corresponds to an individual reasoning operation generated by LLM, and the estimated values serve as process rewards. We instantiate this framework using

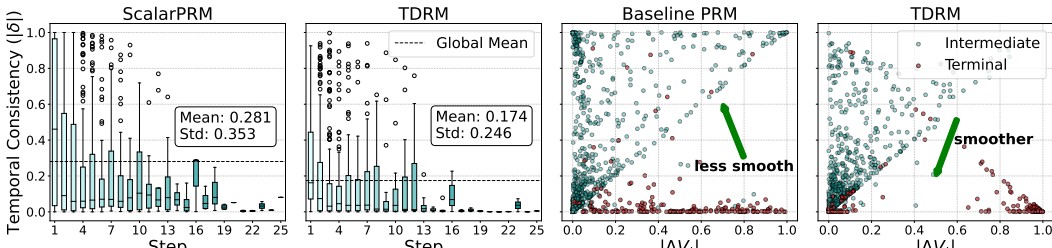

Figure 2: Comparison of reward model smoothness. Left: Box plots of TD error magnitude across reasoning steps (steps segmented by double newlines (i.e., \n\n). TDRM exhibits lower mean and variance of TD errors, indicating smoother and more consistent reward dynamics compared to ScalarPRM —a discriminative process reward model (PRM) that estimates intermediate state values using the Monte Carlo (MC) method, outputting a scalar value (detailed in Appendix Figure 6). Right: Scatter plots of TD error versus value change magnitude. The tighter distribution in TDRM shows a more coherent relationship between error and value updates, especially for intermediate steps, while ScalarPRM exhibits noisier and less structured patterns.

the following $n$-step TD algorithm, where $\phi$ represents the parameters of a PRM, to capture the cumulative impact of intermediate reasoning steps and explicitly model the long-term value:

$$\phi \leftarrow \phi + \left( \underbrace{\sum_{k=0}^{n-1} \gamma^k r_{t+k} + \gamma^n V(s_{t+n}; \phi)}_{\text{TD target}} - V(s_t; \phi) \right) \cdot \nabla_\phi V(s_t; \phi). \quad (2)$$

**Smoothness Analysis.** Smoothness is a crucial property for effective reward modeling in the reasoning process, as it reflects the consistency and stability of value updates across intermediate steps, ensuring that minor changes in reasoning trajectories do not result in disproportionate deviations in value estimation. To measure smoothness, we adopt two complementary approaches to evaluate the behavior of ScalarPRM and our TDRM. *(1) the local Lipschitz constant*, which quantifies the sensitivity of rewards to variations between adjacent states in Table 1 (see details in Appendix F.5). Our analysis shows that TDRM yields a smaller Lipschitz constant on average between consecutive steps, indicating smoother reward transitions and better temporal consistency. *(2) TD error $\delta$* between consecutive reasoning steps and the *value difference $\Delta V_t$* between reasoning steps with Eq. (3) and Eq. (4), providing a combined perspective on assessing the continuity and consistency of the estimated value function. In Figure 2, we compare smoothness by plotting TD error $\delta$ against reasoning steps (steps segmented by double newlines), and TD error $\delta$ vs. value change $|\Delta V_t|$. Here, a step refers to a reasoning segment in the model's generated trajectory, defined by a double newline delimiter. Examining TD error across steps allows us to assess how consistently the reward model evaluates reasoning as the chain progresses. TDRM exhibits lower mean and variance of TD errors (0.174 vs. 0.281) than ScalarPRM, indicating smoother and more stable reward dynamics. In the right panels, each point corresponds to either an intermediate step (**cyan**, reasoning in progress) or a terminal step (**red**, final answer). We distinguish between these because terminal steps are evaluated against the final outcome reward, whereas intermediate steps are judged relative to subsequent reasoning states. A smoother relationship between TD error and value changes at intermediate steps indicates that TDRM provides more coherent trajectory-level reward shaping, while ScalarPRM remains noisier and less structured. These findings afford insights into a stable and consistent reward model design which motivates our TDRM.

Table 1: Lipschitz constant analysis on average.

| Lipz. cont. | ScalarPRM | TDRM |
|---|---|---|
| Avg. ↓ | 0.3331 | 0.2741 |

$$\delta = \left| r + \gamma V(s_{t+1}) - V(s_t) \right|. \quad (3)$$

$$\Delta V_t = \left| V(s_{t+1}) - V(s_t) \right|. \quad (4)$$

$\delta$ measuring the TD error magnitude and $\Delta V_t$ measuring the value change magnitude.

## 3.2 REWARD MODELING

Motivated by the analyses in Section 3.1 as well as insights from prior research [47] which highlights that the length of CoT does not always increase steadily during LLM reasoning, reward shaping emerges as a crucial mechanism for stabilizing the emergent length scaling behavior. In the context of our TD-based PRM, reward shaping serves a dual purpose: it refines the TD updates by providing structured feedback and mitigates the volatility of reward signals across different reasoning lengths.

**Cosine Reward.** To stabilize reasoning length, we leverage the cosine-based reward function [47] that adapts to the correctness of reasoning steps and their relative lengths, assigning distinct reward ranges for correct ($Y = 1$) and incorrect ($Y = 0$) steps, formalized as:

$$r_t = \begin{cases} CosRew(L_{gen}, L_{max}, r_0^c, r_L^c), & if \quad Y = 1 \\ CosRew(L_{gen}, L_{max}, r_0^w, r_L^w), & if \quad Y = 0 \end{cases}. \tag{5}$$

Here, $L_{gen}$ represents the current generation length of the reasoning step, while $L_{max}$ denotes the maximum length across all generated steps. The parameters $r_0^c$ and $r_0^w$ specify the initial rewards for correct and incorrect steps when $L_{gen} = 0$, set to 1 and 0, respectively. Conversely, $r_L^c$ and $r_L^w$ define the terminal rewards at $L_{gen} = L_{max}$, with values of 2 and -10, respectively. The binary label $Y$ serves as a correctness indicator for each step. The cosine reward function itself is defined as:

$$CosRew(l, L, r_{min}, r_{max}) = r_{min} + \frac{1}{2}(r_{max} - r_{min})\left(1 + cos\left(\frac{l\pi}{L}\right)\right). \tag{6}$$

This formulation ensures that the reward begins at its maximum value ($r_{max}$) and gradually decays to the minimum ($r_{min}$) as the reasoning length $l$ approaches the maximum length $L$.

**Temporal Difference (TD).** Once the reward function is defined, we can integrate it with the temporal difference framework to update our PRM. Leveraging the general TD update formula from Eq. (2), we use $r_t$ as the reward in our specific scenario, and set the step size as 1 (referring to the 1-step TD update rule, where the value estimate depends on the immediate reward and the next-step value). We can then derive our TD target with Eq. (2) and Eq. (5):

$$v_t = \sum_{k=0}^{n-1} \gamma^k r_{t+k} + \gamma^n V(s_{t+n}). \tag{7}$$

To align with the desired range of feedback signals, we further process the TD target by clamping it within the interval $[0, 1]$, which yields our final clamped TD target $\tilde{v}_t$:

$$\tilde{v}_t = \begin{cases} V_t, & \text{if } \texttt{is\_terminal}(t) \\ \min\left(\max\left(\sum_{k=0}^{n-1} \gamma^k r_{t+k} + \gamma^n V(s_{t+n}), 0\right), 1\right), & \text{otherwise} \end{cases} \tag{8}$$

In the terminal states, where no subsequent states exist to contribute to the TD calculation, we directly set the target to $V_t$. This integration of the custom reward function with TD learning allows our PRM to effectively capture the temporal dynamics of LLM reasoning, providing more informed and stable guidance for policy optimization. We present a detailed algorithm in Algorithm 3. In Table 5, we set $n$ to each of $\{1, 2, 3\}$ and explore how different $n$ affects PRM performance.

TD learning inherently promotes reward smoothness through its core mechanisms. **First, TD learning estimates the value of states by leveraging both immediate rewards and future value predictions**, which effectively reduces reward fluctuations caused by noisy single-step feedback. **Second, the temporal difference update aligns the value estimates of consecutive states**, fostering consistency in the reward signal across the sequence. For our proposed TDRM, the integration of n-step TD further enhances this smoothness because n-step TD aggregates rewards over multiple steps to mitigate local noise.

Notably, **this mechanism aligns naturally with the autoregressive token-generation process of LLMs**. Since LLMs generate text sequentially (predicting the next token based on previous context), TD learning—by its focus on temporal dependencies between consecutive states—seamlessly adapts to modeling reward signals across the incremental generation process, ensuring smoothness that is inherently compatible with the model's autoregressive nature.

Additionally, **empirical evidence from our analyses in Figures 8 (Reward distribution over different reasoning steps) and Figure 9 (Training reward distribution) supports this claim**. These figures demonstrate that TD-based reward shaping leads to more stable training dynamics for LLMs, with reduced variance in reward signals and more consistent performance improvements across training steps. This further validates that TD learning not only induces smoother rewards but also translates to more effective LLM RL training.

**TD-$\lambda$.** Besides applying $n$-step TD, we also investigate TD-$\lambda$ as an alternative. TD-$\lambda$ generalizes $n$-step TD and functions as an online algorithm that offers greater flexibility. Due to its online nature, TD-$\lambda$ allows PRM to propagate information to earlier states as soon as it observes a reward. For example, in the backward view of TD-$\lambda$, if an intermediate step is incorrect, it can immediately update state values of the preceding states. In contrast, in $n$-step TD, the corresponding states would not receive updates until future episodes. The pseudo-code and results for PRM training using TD-$\lambda$ are shown in Algorithm 2 and Figure 4. Notably, in Algorithm 2, we slightly abuse the notation by writing $V(s_t)$ for the value of model logits instead of the sigmoid values.

**Loss Function.** In optimizing our PRM, we employ Cross-Entropy Loss that leverages a clamped TD target $\tilde{v}_t$ as a soft label for each reasoning step, enabling the model to learn from the temporal consistency of rewards as:

$$\mathcal{L}_{\text{PRM}} = -\mathbb{E}_{\tau_{\text{PRM}} \sim \mathcal{D}_{\text{PRM}}} \left[ \frac{1}{|\tau_{\text{PRM}}|} \sum_{t=1}^{|\tau_{\text{PRM}}|} \tilde{v}_t \log(p_t) + (1 - \tilde{v}_t) \log(1 - p_t) \right], \tag{9}$$

where $\tau_{\text{PRM}} = \{(s_1, r_1), \ldots, (s_T, r_T)\}$ is the trajectory containing each step and the corresponding reward $r_t$, and $p_t$ refers to the model's output probability at step $t$, derived by applying the sigmoid function to the output logits. In practice, reasoning steps from diverse trajectories are randomly batched to facilitate minibatch training, ensuring the loss function captures both local step-wise rewards and global trajectory dynamics.

### 3.3 ONLINE REINFORCEMENT LEARNING

Our algorithm operates online, dynamically calculating TD targets using state values on-the-fly during training. Unlike offline algorithms that rely on pre-computed state values, TDRM adapts to evolving trajectories, leveraging seen trajectories to estimate state values for unseen ones. This adaptability improves value prediction accuracy and enhances the consistency and robustness of the reward model.

**Verifiable Reward.** In our RL training, we follow the verifiable reward $R_{\text{verifiable}}$ used in R1 [5]. $R_{\text{verifiable}}$ is defined as a function that checks the format of the predicted answer $\hat{g}$ (`has_boxed`) and assesses the equivalence between the prediction $\hat{g}$ and the ground-truth $g$ (`is_equivalent`):

$$R_{\text{verifiable}}(\hat{g}, g) = \begin{cases} 1, & \text{if } \texttt{is\_equivalent}(\hat{g}, g) \text{ and } \texttt{has\_boxed}(\hat{g}) \\ 0, & \text{if } \neg\texttt{is\_equivalent}(\hat{g}, g) \text{ and } \texttt{has\_boxed}(\hat{g}) \\ -1, & \text{otherwise} \end{cases} \tag{10}$$

While $R_{\text{verifiable}}$ is straightforward and interpretable, it considers only the end answer and omits assessment of intermediate reasoning steps. A more detailed explanation of `is_equivalent` and `has_boxed` can be found in Appendix F.4.

**Process-based Reward.** Rule-based verifiable rewards often encounter a critical limitation: they assign identical rewards to trajectories that produce correct answers via incorrect intermediate steps. To address this gap and capture the temporal dynamics of reasoning, our PRM plays a pivotal role in online RL. By assigning rewards to intermediate states based on their estimated values, PRM provides a more fine-grained feedback signal, effectively mitigating the "right answer, wrong process" issue. Specifically, the process-based reward at step $t$ is defined as the state value output by the PRM through $R_{\text{PRM}}(s_t) := \text{PRM}_\phi(s_t)$. PRM$_\phi$ denotes our proposed PRM, parameterized by $\phi$.

**Effective Combination for RL.** In TDRM, we harness the complementary strengths of verifiable and process-based rewards through a linear combination, enabling a more comprehensive and nuanced reward signal for online RL. The final reward function $R_{\text{final}}$ is formulated as:

$$r_{\text{final}} = a r_{\text{PRM}} + (1 - a) r_{\text{verifiable}}, \tag{11}$$

where the hyper-parameter $a$ balances the influence of process-based feedback against outcome-based verification. Finally, this combined reward $r_{\text{final}}$ as $r_i$ is used in $\hat{A}_{i,j} = \frac{r_i - \text{mean}(\{r_1, r_2, ..., r_G\})}{\text{std}(\{r_1, r_2, ..., r_G\})}$ to train the GRPO objective, enhancing the overall performance and data efficiency of the learning process.

**Algorithm Implementation.** As presented in Algorithm 1, we outline the overall training process of TDRM for integrating verifiable and process-based rewards in online RL. Additionally, Algorithm 3 provides a step-by-step breakdown of the $n$-step TD method used for training PRM.

---

**Algorithm 1:** Process of TDRM

---

**Notation:** GRPO: group relative policy optimization; $\text{PRM}_\phi(s_t)$: PRM logits for step $s_t$
**Input:** Initial policy model $\pi_\theta$; process reward model $\text{PRM}_\phi$; verifiable reward function $R_{\text{verifiable}}$; task prompts $\mathcal{D}_{\text{policy}}$; final reward $R_{\text{final}}$; hyperparameters $\alpha$
1: Reference model $\pi_{\text{ref}} \leftarrow \pi_\theta$
2: **for** Iteration $= 1, \ldots, I$ **do**
3:     Sample a mini-batch $\mathcal{D}_b$ from $\mathcal{D}_{\text{policy}}$
4:     Set old policy $\pi_{\text{old}} \leftarrow \pi_\theta$
5:     Sample $G$ trajectories $\{\tau_i\}_{i=1}^G$ from $\pi_{\text{old}}$ for each question $q \in \mathcal{D}_b$
6:     **for** each trajectory $\tau_i = \{s_1, \ldots, s_T\}$ **do**
7:         Compute verifiable reward $r_{\text{verifiable}}^{(\tau_i)}$ for $\tau_i$ through Eq. (10)
8:         Compute process-based reward $r_{\text{PRM}}^{(s_{T-1})} \leftarrow \text{PRM}_\phi(s_{T-1})$ for $s_{T-1}$ through $R_{\text{PRM}}(s_t) := \text{PRM}_\phi(s_t)$
9:         Compute final reward $r_{\text{final}}^{(\tau_i)} \leftarrow a \cdot r_{\text{verifiable}}^{(\tau_i)} + (1-a) \cdot r_{\text{PRM}}(s_{T-1})$ for $\tau_i$ through Eq. (11)
10:     **end for**
11:     Compute advantages $\hat{A}_{i,j}$ for the $j$-th token of each $\tau_i$ using group relative advantage estimation
12:     Update the policy $\pi_\theta$ through maximizing the GRPO objective using $\hat{A}_{i,j}$
13: **end for**
**Output:** Optimized policy $\pi_\theta$

---

## 4 EXPERIMENTS

In this section, we benchmark TDRM in two scenarios, i.e., (1) inference-time verification and (2) training-time online reinforcement learning.

### 4.1 EXPERIMENTAL SETTINGS

**Evaluation Metrics and Benchmarks.** *(1) For inference-time verification*, we compare different reward models under two key settings. *Best-of-$N$ Sampling* works by first generating a pool of $N$ potential outputs and selecting the best candidate using the RM. We test with $N \in \{128, 1024\}$ and evaluate on GSM8K [4] and MATH-500 [11]. *Greedy Search* [21] generates outputs by iteratively selecting the highest-scoring sequences. To improve exploration, the branching factor is set to $m \in \{2, 4, 8, 16\}$, and experiments are performed on MATH-500. For a fair comparison, pre-generate reasoning trajectories are utilized during inference. Accuracy is used as the evaluation metric for both strategies (see more details in Appendix F.1). *(2) For training-time online RL*, we benchmark TDRM against leading methods on five difficult datasets: MATH-500, Minerva Math [19], Olympiad Bench [10], AIME24, and AMC23. Following SimpleRL [50], we evaluate performance using the Pass@1 metric with greedy decoding.

### 4.2 MAIN EXPERIMENTAL RESULTS

**Reward Modeling and Inference Scaling Results.** Corresponding Section 2.2, Figure 6 provides the definition comparison and MATH-500 results of recent reward models. Table 2 and 3 present the results of Best-of-$N$ sampling across different models and datasets, providing empirical evidence of TDRM's superiority. Firstly, TDRM outperforms ScalarPRM and ScalarORM on the MATH-500 dataset as the sampling budget increases from Best-of-128 to Best-of-1024. Specifically, with DS-R1-Distill-Qwen-7B, TDRM achieves relative improvements of 6.7% over ScalarORM and 3.9% over ScalarPRM;

Table 2: Results on MATH-500 using the RM for selection in Best-of-$N$ sampling. The PRM backbone is DeepSeek-R1-Distill-Qwen-7B.

| Method | DS-R1-Distill-Qwen-7B | | Llama3.1-8B-Instruct | |
|---|---|---|---|---|
| | Best-of-128 | Best-of-1024 | Best-of-128 | Best-of-1024 |
| ScalarORM | 52.0 | 54.8 | 42.2 | 42.8 |
| ScalarPRM | 53.4 | 56.2 | **44.4** | 44.8 |
| TDRM | **54.2** | **58.4** | 43.2 | **45.6** |

Table 3: Results on GSM8K in Best-of-128 sampling.

| Method | Result |
|---|---|
| ScalarORM | 69.29 |
| ScalarPRM | 71.34 |
| TDRM | **73.24** |

with Llama3.1-8B-Instruct, the respective relative gains are 6.5% compared to ScalarORM and 1.8% compared to ScalarPRM. This strongly indicates that TDRM is more reliable and can consistently identify the best responses with larger sampling budgets. Notably, the GSM8K results in Table 3 utilize samples generated by `Mistral-7B-Instruct-v0.2`, which is different from the reward models' training data, demonstrating TDRM's superior ability to generalize to new data distributions.

In tree search evaluations, as shown in Figure 3, TDRM again demonstrates superior performance with Qwen2.5-Math-7B and provides a more accurate verification of reasoning trajectories. Moreover, TDRM exhibits enhanced reliability, with its accuracy improving as the number of search branching factors increases from 2 to 16, indicating its effectiveness in navigating complex decision spaces. In addition, as shown in Figure 7, TDRM further validates its ability on unseen data distributions (i.e., Mistral data) compared to baseline methods.

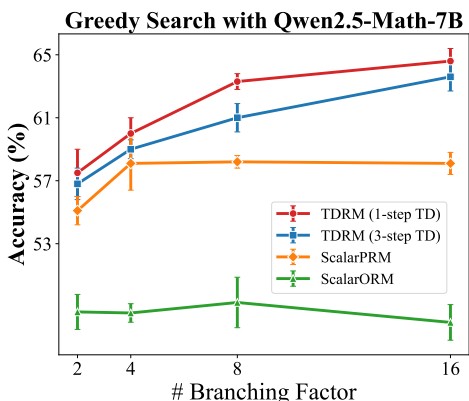

Figure 3: Comparison of TDRM versus baselines on Greedy Search, using Qwen2.5-Math-7B as the backbone.

**Online Reinforcement Learning Results.** Table 4 compares the RL training outcomes of TDRM against the state-of-the-art methods, demonstrating its superiority across 8 model variants (5 series) using only 2.5k MATH Level-3 prompts. Spanning diverse model sizes and pre-training paradigms, TDRM consistently achieves the highest average accuracy, underscoring its reliability in RL training. For example, TDRM beats all the other methods — whether using verifiable rewards or reward models — by an average of 0.9% to 4.4% over the second-best model, with a notable 36.7% Pass@1 on AIME24 on Qwen2.5-Math-7B, highlighting its significant advancement in mathematical reasoning. Notably, on smaller Qwen2.5-(0.5B, 1.5B), which exhibit weaker inherent math capabilities, training with ScalarPRM or ScalarORM alone leads to model collapse. In contrast, TDRM 's linear combination of verifiable and process-based rewards ensures stable performance and superior data efficiency, enabling consistent learning even with limited training samples.

Table 5: Results of $n$-step TD on MATH-500 when backbone is DS-R1-Distill-Qwen-7B.

| $n$ | Best-of-128 | Best-of-1024 |
|---|---|---|
| 1 | 54.2 | **58.4** |
| 2 | **55.4** | 56.2 |
| 3 | 54.2 | 56.8 |

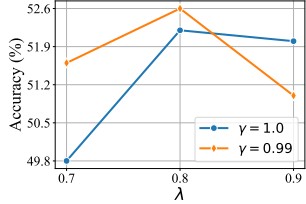

Figure 4: TD-$\lambda$ results on MATH-500 in Best-of-128 sampling.

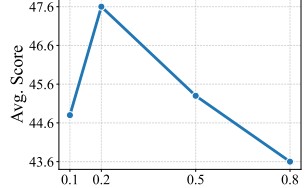

Figure 5: Avg. performance of online RL training vs $a$ on Qwen2.5-Math-7B.

### 4.3 ANALYSIS AND ABLATION STUDIES

We study the features of TDRM through comprehensive analyses: reward distribution comparison, varying lookahead steps in TDRM, TD-$\lambda$, and the tradeoff between verifiable and process rewards.

Table 4: Evaluation results on standard mathematical benchmarks under a constrained data system of 2.5k samples. We highlight the top score in **bold** and the second-best by underlining it. The relative improvement (**%Improv.**) for each method is computed based on the performance in this setup.

| Model | Data Size | MATH 500 | Minerva Math | Olympiad Bench | AIME24 (Pass@1) | AMC23 | Avg. |
|---|---|---|---|---|---|---|---|
| **Backbone is Base Model, Qwen Series** | | | | | | | |
| **Qwen2.5-0.5B** | - | 15.8 | 4.8 | 2.8 | 0.0 | 12.5 | 7.2 |
| + SimpleRL (Greedy) | 50.1k | 32.6 | 8.1 | 9.0 | 0.0 | 15.0 | 12.9 |
| + ScalarPRM | | 3.4 | 2.2 | 1.9 | 0.0 | 5.0 | 2.5 |
| + ScalarORM | 2.5k | 6.2 | 2.2 | 2.8 | 0.0 | 5.0 | 3.2 |
| + Rule-based | | 29.8 | 4.0 | 7.0 | 0.0 | 12.5 | 10.7 |
| + Ours | | 26.2 | 4.8 | 7.1 | 0.0 | 15.0 | **10.8** (+0.9%) |
| **Qwen2.5-1.5B** | - | 29.6 | 6.6 | 6.5 | 0.0 | 12.5 | 11.0 |
| + SimpleRL (Greedy) | 50.1k | 22.6 | 2.6 | 8.4 | 0.0 | 5.0 | 7.7 |
| + ScalarPRM | | 0.2 | 0.0 | 0.0 | 0.0 | 0.0 | 0.0 |
| + ScalarORM | 2.5k | 1.8 | 0.0 | 0.6 | 0.0 | 5.0 | 1.5 |
| + Rule-based | | 58.0 | 12.1 | 18.7 | 0.0 | 27.5 | 23.3 |
| + Ours | | 52.8 | 9.9 | 17.8 | 3.3 | 35.0 | **23.8** (+2.1%) |
| **Backbone is Chat Model, GLM Series** | | | | | | | |
| **GLM4-9B-0414** | - | 65.8 | 36.8 | 28.7 | 10.0 | 42.5 | 36.8 |
| + ScalarPRM | | 67.0 | 38.6 | 31.9 | 6.7 | 45.0 | 37.8 |
| + ScalarORM | 2.5k | 68.2 | 39.3 | 30.2 | 10.0 | 42.5 | 38.0 |
| + Rule-based | | 72.8 | 37.5 | 37.0 | 16.7 | 40.0 | 40.8 |
| + Ours | | 72.2 | 37.1 | 32.0 | 20.0 | 47.5 | **41.8** (+2.5%) |
| **Backbone is Reasoning Model, GLM Series** | | | | | | | |
| **GLM-Z1-9B-0414** | - | 93.6 | 43.8 | 65.5 | 73.3 | 92.5 | 73.7 |
| + ScalarPRM | | 94.0 | 47.8 | 66.4 | 76.7 | 92.5 | 75.5 |
| + ScalarORM | 2.5k | 95.0 | 46.7 | 65.9 | 76.7 | 97.5 | 76.4 |
| + Rule-based | | 95.6 | 43.4 | 65.2 | 73.3 | 97.5 | 75.0 |
| + Ours | | 94.6 | 44.9 | 66.5 | 80.0 | 97.5 | **76.7** (+0.4%) |
| **Backbone is Base Model, Qwen-Math Series** | | | | | | | |
| **Qwen2.5-Math-1.5B** | - | 42.2 | 8.8 | 27.0 | 10.0 | 37.5 | 25.1 |
| + SimpleRL (Greedy) | 50.1k | 59.8 | 13.6 | 29.9 | 10.0 | 37.5 | 30.2 |
| + ScalarPRM | | 66.2 | 17.3 | 28.7 | 13.3 | 50.0 | 35.1 |
| + ScalarORM | 2.5k | 41.6 | 8.5 | 27.0 | 10.0 | 40.0 | 25.4 |
| + Rule-based | | 67.6 | 21.3 | 31.0 | 6.7 | 52.5 | 35.8 |
| + Ours | | 66.2 | 18.4 | 30.1 | 13.3 | 55.0 | **36.6** (+2.2%) |
| **Qwen2.5-Math-7B** | - | 63.6 | 12.5 | 25.8 | 13.3 | 42.5 | 31.5 |
| + Our Template | - | 68.8 | 16.2 | 31.1 | 13.3 | 62.5 | 38.4 |
| + SimpleRL-Zero | 8.5k | 77.8 | 31.2 | 37.5 | 23.3 | 62.5 | 46.5 |
| + SimpleRL (Greedy) | 50.1k | 78.2 | 27.6 | 40.3 | 26.7 | 60.2 | 46.6 |
| PPO | - | 72.2 | 32.0 | 35.9 | 26.7 | 55.0 | 44.4 |
| GRPO | - | 77.8 | 39.7 | 39.1 | 20.0 | 57.5 | 46.8 |
| Dr. GRPO† | - | 74.6 | 30.1 | 37.3 | 26.7 | 50.0 | 43.7 |
| OpenReasoner-Zero | - | 82.4 | 31.6 | 47.9 | 13.3 | 54.2 | 45.9 |
| + ScalarPRM | | 75.8 | 29.0 | 36.4 | 26.7 | 60.0 | 45.6 |
| + ScalarORM | 2.5k | 71.2 | 22.1 | 37.5 | 20.0 | 50.0 | 40.2 |
| + Rule-based | | 73.2 | 25.0 | 37.8 | 23.3 | 65.0 | 44.9 |
| + Ours | | 74.6 | 26.8 | 37.3 | 36.7 | 62.5 | **47.6** (+4.4%) |
| **Backbone is Reasoning Model, DeepSeek Series** | | | | | | | |
| **DS-R1-Distill-Qwen-1.5B** | - | 70.6 | 26.5 | 32.1 | 16.7 | 50.0 | 39.2 |
| + ScalarPRM | | 74.2 | 29.0 | 35.7 | 33.3 | 60.0 | 46.4 |
| + ScalarORM | 2.5k | 77.4 | 30.5 | 38.5 | 33.3 | 60.0 | 47.9 |
| + Rule-based | | 75.4 | 26.8 | 36.1 | 20.0 | 57.5 | 43.2 |
| + Ours | | 79.8 | 30.5 | 38.2 | 30.0 | 70.0 | **49.7** (+3.8%) |
| **DS-R1-Distill-Qwen-7B** | - | 88.0 | 43.0 | 49.9 | 63.3 | 82.5 | 65.3 |
| SEED-GRPO | 8.5k | 91.6 | 38.6 | 61.5 | 50.0 | 78.3 | 64.0 |
| + ScalarPRM | | 87.6 | 50.7 | 49.8 | 53.3 | 85.0 | 65.3 |
| + ScalarORM | 2.5k | 90.4 | 50.7 | 52.7 | 43.3 | 90.0 | 65.4 |
| + Rule-based | | 89.6 | 46.0 | 52.4 | 50.0 | 82.5 | 64.1 |
| + Ours | | 91.8 | 50.4 | 54.1 | 53.3 | 87.5 | **67.4** (+3.0%) |

Given that TDRM (3-step TD) is the primary configuration for RL training, the following studies focus on this setup, with ScalarPRM serving as the default comparator unless specified otherwise.

Table 6: TD-learned value function comparison between ScalarPRM and our TDRM.

| Metric | ScalarPRM | TDRM |
|---|---|---|
| Point-Biserial Correlation with Final Accuracy | 0.148 | 0.195 |
| Pearson Correlation with Step-wise Labels | -0.061 | 0.018 |

**TD-learned Value Function Comparison.** To validate our TD-learned reward signals, we utilize two alternative evaluation metrics that directly reflect the superiority of TDRM 's value function beyond TD error statistics: *1) correlation with final task accuracy (a pragmatic proxy for value function quality)*, and *2) alignment with human-annotated step-wise correctness (ground truth for process rewards)*. Regarding 1, for complex reasoning tasks, a high-quality value function should predict the likelihood of a trajectory leading to a correct final output. We compute the point-biserial correlation between the model's estimated value of the initial step ($s_0$) and the final task accuracy of the full trajectory. This metric evaluates the value function's ability to "foresee" trajectory success, a critical property for guiding RL training and inference search. Again, TDRM outperforms ScalarPRM by 4.7%, a 31.8% stronger correlation, confirming that its value function provides more meaningful signals for predicting trajectory quality—complementing the TD error analysis with a pragmatic, task-relevant metric. Regarding 2), we compute Pearson correlation between each model's step-wise value estimates and human correctness labels from RLHFlow/Mistral-PRM-Data. TDRM achieves a significantly higher correlation (7.9% points higher than ScalarPRM), but more importantly, **TDRM maintains positive correlation with ground truth (+0.018) while ScalarPRM shows negative correlation (-0.061)**. This indicates that TDRM's value function is fundamentally better aligned with ground-truth reasoning quality, whereas ScalarPRM's signals are actually misaligned with the ground truth **(both correlations statistically significant: p < 0.001).**

***n*-step TD.** To study the effect of look-ahead steps $n$, we present the Best-of-$N$ results for TDRM trained with 1, 2, and 3-step TD in Table 5. While 1-step TD performs the best under a larger sampling budget (Best-of-1024), 2-step TD achieves the best under a smaller number of candidates (Best-of-128). This suggests that a moderate lookahead step may help improve sample efficiency, while shorter horizons exhibit greater robustness and generalize better with a larger budget.

**TD-$\lambda$.** Building on the $n$-step TD framework, where TD-$\lambda$ provides a mechanism to balance between TD(0) and TD(1), we evaluate TD-$\lambda$ under varying values of $\lambda$ and different discount factor $\gamma$. As shown in Figure 4, the interaction between $\lambda$ and $\gamma$ has a significant non-linear impact on model accuracy. Specifically, $\lambda = 0.8$ consistently achieves the highest accuracy for both discount factors. However, as $\lambda$ increases to 0.9, the accuracy declines. These results highlight that tuning of $\lambda$ around 0.8 is critical for balancing temporal consistency and achieving optimal performance.

**Reward Combination Tradeoff.** In the RL training of TDRM, we analyze the linear combination of the process reward and the verifiable reward via the coefficient $a$ (verifiable reward coefficient: $1 - a$). As shown in Figure 5, performance peaks at $a = 0.2$, with significant degradation for both higher and lower values. This indicates process rewards serve best as a complementary signal—too low a weight introduces insufficient guidance, while excessive weight amplifies noise.

## 5 CONCLUSION

TDRM tackles the challenge of temporal inconsistency in reward models by introducing TD regularization, which enhances reward density and stability. Across Best-of-$N$ and tree-search scenarios, TDRM-trained PRMs consistently improve performance and complement verifiable reward methods, enabling more data-efficient RL training and stronger LLM policies on 8 models.

These results show that incorporating temporal consistency into reward models not only stabilizes RL training but also opens the door to more scalable RLHF pipelines, higher-quality inference-time search, and broader applications in aligning LLMs with complex objectives.

REPRODUCIBILITY STATEMENT

We provide pseudo-code in the Algorithm 1, Algorithm 2, and Algorithm 3 for TDRM training process, TD-$\lambda$, and $n$-step TD for PRM training. We provide experimental settings in Section 4.1 and F.1. We also release all code to promote reproducibility.

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

## A  STATEMENT OF LLM USAGE

This manuscript was prepared by the authors, who take full responsibility for its content. Large language models (ChatGPT, etc.) were used solely for language polishing and grammar suggestions. No generated text or analysis was included without human verification.

## B  TABLE OF NOTATIONS

Table 7: Table of Notations

| Symbol | Meaning |
|---|---|
| $N$ | number of generations for Best-of-$N$ |
| $\mathcal{S}$ | state space |
| $\mathcal{A}$ | action space |
| $f$ | transition function |
| $R$ | reward function ($R: \mathcal{S} \times \mathcal{A} \to r, r \in \mathbb{R}$) |
| $s_t$ | state (token sequence or context for LLM) at step t |
| $a_t$ | action (newly generated token) at step t |
| $(x_0, \ldots, x_L)$ | input prompt |
| $(y_0, \ldots, y_{t-1})$ | sequence of generation tokens |
| $T$ | terminal step |
| $\rho_\pi$ | trajectory distribution |
| $\beta$ | KL coefficient |
| $q$ | query |
| $O = o_1 \ldots o_G$ | response |
| $r_t$ | reward at step $t$ |
| $n$ | $n$-step TD |
| $\gamma$ | discount factor |
| $V$ | value function |
| $\alpha$ | step size in TD |
| $v_t$ | TD target |
| $\tilde{v}_t$ | clamped TD target |
| $V_t$ | TD target at terminal step |
| $\hat{g}$ | predicted answer |
| $g$ | ground truth answer |
| $R_{\text{verifiable}}$ | verifiable reward function |
| $R_{\text{PRM}}$ | PRM reward function |
| $\text{PRM}_\phi$ | PRM, $\phi$ refers to the parameter |
| $a$ | coefficient for "TDRL" |
| $\hat{A}_{i,j}$ | advantage for the $j$-th token of each $\tau_i$ using group relative advantage estimation |
| $\tau_i$ | in Alg. 1, the $i$-th trajectory in batch $\mathcal{D}_b$ sampled from task prompts $\mathcal{D}_{\text{policy}}$ |
| $\tau_{\text{PRM}}$ | trajectory in TD PRM dataset $\mathcal{D}_{\text{PRM}}$ |
| $U$ | discounted return |

## C  RELATED WORK

### C.1  REASONING PROCESS REWARD

LLMs have achieved significant performance improvement in advanced complex reasoning scenarios [14; 29; 20] through step-by-step reasoning. For example, CoT [42], ToT [46], SFS [22], and MCTS [52] have progressed in reasoning tasks by analyzing complex questions and providing guidance for models to obtain correct solutions. Uesato [40] and Light et al. [23] propose the ORM that detects the final result and PRM that provides the feedback for intermediate reasoning steps, and demonstrate that PRM is more effective than ORM for obtaining correct step-level process and avoiding the false positive steps that match with final correct answer with incorrect solutions.

Step-DPO [18] checks step-by-step answers and collects positive and negative step-level solutions for training direct preference optimization [30] rather than evaluating the correctness of whole solutions and final answer. Math-shepherd [41] and ReST-MCTS* [52] introduce reinforced training by integrating process reward with tree search to collect high-quality reasoning paths for LLMs in mathematical reasoning. Despite their effectiveness, the research to obtain automated, more correct, and label-free process rewards remains unexplored. To implement this goal, we propose a new reward function for process reward optimization and online training of LLMs.

## C.2 Reinforcement Learning Training of LLMs

Direct preference optimization (DPO) [30] optimizes models by learning positive and negative pairs. Compared to DPO, Proximal Policy Optimization (PPO) [31] is an effective online RLHF algorithm but requires high GPU memory and is challenged in real-use scenarios. To fill the gap, reinforce leave-one-out (RLOO) [2] is proposed to load the policy, reference, and reward models to memory, and model the entire completed token as a single action.

## C.3 Temporal Difference in Reinforcement Learning

TD plays a vital role in connecting model-based and model-free methods within RL, estimating state values by merging immediate rewards with discounted future state values. The foundational 1-step TD algorithm [35] updates state value estimates using TD errors, enabling agents to learn optimal policies online. TD methods have also been integrated with policy search techniques, resulting in TD-based policy gradient algorithms such as A2C [27] that leverage TD errors to optimize policies, achieving great success in game playing [37; 38]. In the deep RL realm, DQN [26] and its variants utilize TD learning to train neural networks approximating the Q-function. Therefore, TD learning is a natural method for training reliable and smoother reward models for RL training.

## D Detailed comparison with related methods

We present conceptual distinctions and discussions with related approaches: classical actor-critic methods, also use TD learning or value-based methods, ScalarPRM, and reward models.

### D.1 Comparison with classical actor-critic methods

In the context of LLM fine-tuning literature (e.g., ReST-MCTS* [52], TS-LLM [7], pDPO [17]), reward models and classical value functions share foundational similarities—both aim to quantify the value of states/trajectories, and both can be trained via TD learning. However, our proposed TDRM, while drawing on these classical RL principles, is tailored to **address the unique challenges of LLM training and test-time inference**, leading to meaningful conceptual and practical differences beyond the offline-online distinction you noted:

**(1) Alignment with LLM-specific scale and extended reasoning processes:** Classical actor-critic methods are designed for environments with relatively compact state spaces and short interaction trajectories. In contrast, our TDRM is optimized for LLMs' characteristics—**massive model scales, long sequential token-generation trajectories** (often tens to hundreds of steps), and the need to model reasoning processes that unfold over extended contexts. This requires adaptations like handling long-range temporal dependencies in reward signals and ensuring computational feasibility for offline training on large-scale LLM-generated data.

**(2) Offline training with RL training and inference-time utilities:** While classical critics are learned online alongside the actor to guide immediate policy updates, our TDRM is trained offline on a large corpus of pre-generated trajectories. Critically, this offline training enables the reward model to serve a dual purpose: not only **guiding offline RL training** (as a substitute for online critics) but also **supporting inference-time search** (e.g., tree-based search or Best-of-$N$) by providing reliable reward estimates for candidate trajectories. This latter capability is particularly vital in LLM literature, as Inference-time search can explore multiple candidate trajectories, then select the highest-quality output to enhance reasoning performance [52; 14] while avoiding the prohibitive costs of retraining. Notably, this versatility is rarely emphasized in classical actor-critic frameworks, where critics are

Table 8: Discussions with prior methods that also use TD learning. S: Only supports specific tasks (for example, $\mathcal{B}$-Coder is only compatible with code generation and does not support general inference); -: Indirect support or no explicit optimization (for example, Q-SFT can handle long sequences but is not designed or optimized for temporal consistency. In contrast, we implemented reward shaping optimization for the reward $r$ in TD); ✗: unsupport; ✓: support;

| Core Features | TDRM | Q-sft [12] | DQO [15] | $\mathcal{B}$-Coder [49] | ILQL [33] |
|---|---|---|---|---|---|
| **Reward/Value Paradigm** | | | | | |
| - Trains a decoupled PRM | ✓ | ✗ | ✗ | ✗ | ✗ |
| - Directly optimizes Q-function/policy | ✗ | ✓ | ✓ | ✓ | - |
| - Uses n-step TD/TD-$\lambda$ for training | ✓ | ✗ | ✗ | ✗ | ✗ |
| **Key Design Goals** | | | | | |
| - Prioritizes temporal consistency of rewards | ✓ | ✗ | ✗ | ✗ | ✗ |
| - Conducts inference-time search (e.g., BoN, tree) | ✓ | ✗ | ✗ | ✗ | ✗ |
| - Focuses on long-horizon autoregressive generation | ✓ | - | - | S | - |
| **LLM Adaptations** | | | | | |
| - Multi-task generalizability (math, agents) | ✓ | ✓ | ✓ | S | ✓ |
| - Step-wise reward signaling for seq. reasoning | ✓ | - | ✓ | S | ✗ |
| **Training Paradigm** | | | | | |
| - Offline training of reward model | ✓ | ✗ | ✗ | ✗ | ✓ |
| - Conducts STF/RL training | RL | SFT | SFT | SFT | RL |
| - Implicit value estimation | ✗ | ✗ | ✗ | ✗ | ✓ |
| **Task Scope** | | | | | |
| - General LLM RL (math, agents) | ✓ | ✓ | ✓ | S | ✓ |
| - Specialized task (e.g., program synthesis) | ✗ | ✗ | ✗ | ✓ | ✗ |

tightly coupled to online policy learning and not designed to provide the decoupled, reliable reward signals needed for inference-time exploration.

**(3) Adaptation to LLM-specific long-horizon, multi-task reasoning demands**: Classical value functions—even those trained via TD—are designed for single-task, short-horizon environments (e.g., game playing, robotic control) where trajectories rarely exceed dozens of steps and task distributions are narrow. In contrast, our TDRM is tailored to the unique constraints of LLM training: LLMs tackle various distinct tasks (e.g., mathematical reasoning, logical deduction, open-ended generation) across massive, diverse datasets, with autoregressive trajectories often spanning hundreds of tokens per task. TDRM achieves this by leveraging TD learning to model temporal dependencies rather than optimizing for task-specific value accuracy.

## D.2   COMPARISON WITH VALUE-BASED METHODS (Q-LEARNING)

We summarize the core differences between TDRM and value-based methods in Table 8:

**(1) Focus on PRM vs. Policy or Q-Function:** TDRM is a decoupled process reward model trained via TD to induce smooth, step-wise rewards—whereas [12; 15; 49; 33] optimize Q-functions/policies directly (no PRM for sequential reasoning).

**(2) TD for Reward Smoothness:** TDRM leverages n-step TD (also performs TD-$\lambda$) to prioritize temporal consistency (critical for LLM long-horizon generation)—a goal not addressed by (SFT) [12], (Q-optimization) [15], (code-specific SFT) [49], or (implicit Q-learning) [33].

**(3) Inference-Time Utility:** TDRM conducts LLM inference search (e.g., Best-of-$N$, tree search) via reliable reward estimates—unique compared to [12; 15; 49; 33], which focus solely on policy training.

This comparison underscores TDRM 's novel positioning: adapting TD learning to PRM training for LLM-specific challenges (long sequences, sequential reasoning, inference search) that are not addressed by prior value-based RL for language.

### D.3 COMPARISON WITH SCALARPRM

ScalarPRM is a discriminative process reward model (PRM) that estimates the value of intermediate states using the Monte Carlo (MC) method, with the output being a scalar value. Importantly, **ScalarPRM does not leverage TD learning**—its value estimation relies solely on MC, which computes returns by averaging the final rewards of complete trajectories. This stands in contrast to our TDRM, which is **the first work to adopt TD learning for training process reward models (PRMs) and apply the resulting model to LLM RL fine-tuning**.

The core differences between ScalarPRM and our TDRM can be summarized as follows:

**(1) Value estimation method:** ScalarPRM uses MC (trajectory-complete returns), while TDRM uses TD learning (temporal difference updates leveraging immediate rewards and future value predictions).

**(2) Focus on temporal consistency:** ScalarPRM prioritizes scalar value estimation for intermediate states but does not explicitly optimize for the temporal consistency of reward signals. In contrast, TDRM 's TD-based training inherently induces temporally consistent rewards, which we validate by showing that TDRM exhibits a lower mean compared to ScalarPRM.

**(3) Adaptation to LLM sequential generation:** TD learning's emphasis on consecutive state dependencies aligns with LLMs' autoregressive generation, whereas MC's reliance on full trajectory returns makes it less suitable for capturing incremental reasoning progress in long sequential outputs.

### D.4 COMPARISON OF REWARD MODELS

Figure 6 provides a comprehensive comparison of recent reward models from various perspectives (e.g., value type, reward model, value estimation, temporal consistency, and MATH-500 results).

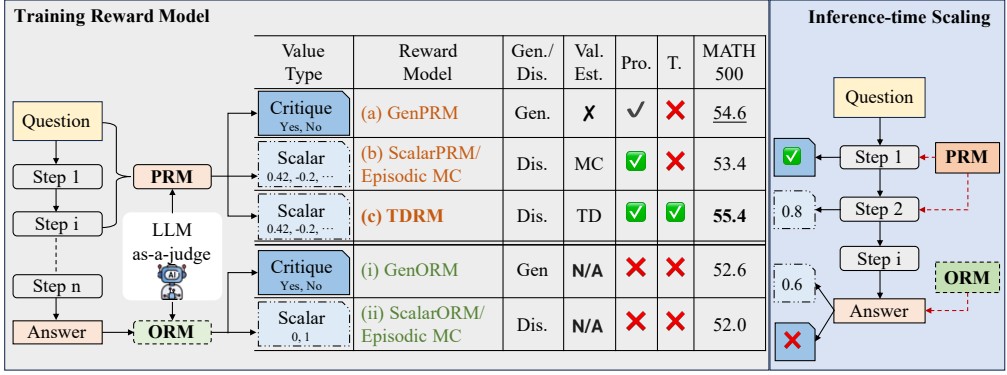

Figure 6: Comparison of recent reward models and TDRM. Gen. and Dis. denote Generative and Discriminative. Val. Est. denotes the method of value estimation. Pro. and T. denote process and temporality. MC and TD denote Monte Carlo and temporal difference.

## E   ALGORITHM DETAILS

We present a backward view of TD-$\lambda$ for PRM training in Algorithm 2 and $n$-step TD for PRM training in Algorithm 3.

## F   EXPERIMENT DETAILS

### F.1   EXPERIMENT SETTINGS

**Evaluation Metrics**  Best-of-$N$ Sampling is designed to balance diversity and optimality across outputs. In our experiments, $N$ is set to $\{128, 1024\}$. While Greedy Search [21] is efficient, it risks suboptimal results due to locally optimal decisions. For a fair comparison and a more thorough study, we pre-generate reasoning trajectories using Mistral-7B-Instruct-v0.2 for GSM8K (128 outputs

---

**Algorithm 2:** Backward view of TD-$\lambda$ for PRM training

---

**Notation:** $s_t$: state; $a_t$: action; $r_t$: reward; $V(s_t)$: state value; $\hat{V}(s_t)$: updated value estimate; $e(s)$: eligibility trace; $\delta$: TD error; $\pi$: policy
**Input:** Dataset $\mathcal{D}_{\text{PRM}}$ of trajectories with rewards $\{r_t\}_{t=1}^{T}$; process reward model $\text{PRM}_\phi$ with parameters $\phi$; discount factor $\gamma$; step size $n$; eligibility trace decay rate $\lambda$

1: Initialize total loss: $\mathcal{L} \leftarrow 0$
2: **for** each trajectory $\tau = \{(s_1, r_1), \ldots, (s_T, r_T)\}$ in $\mathcal{D}_{\text{PRM}}$ **do**
3:     Initialize value estimates: $\hat{V}(s_t) \leftarrow \text{PRM}_\phi(s_t), \forall s_t$
4:     Initialize eligibility traces: $e(s_t) \leftarrow 0, \forall s_t$
5:     **for** $t = 1$ to $T - 1$ **do**
6:         $V(s_{t+1}) \leftarrow \text{PRM}_\phi(s_{t+1})$
7:         $e(s_t) \leftarrow \gamma\lambda \cdot e(s_t)$
8:         $e(s_t) \leftarrow e(s_t) + 1$
9:         $\delta \leftarrow r_t + \gamma \cdot \hat{V}(s_{t+1}) - \hat{V}(s_t)$
10:         **for** $j = 1$ to $t$ **do**
11:             $\hat{V}(s_j) \leftarrow \hat{V}(s_j) + \delta \cdot e(s_t)$
12:         **end for**
13:     **end for**

14:     $\mathcal{L} \leftarrow \sum_{t=1}^{T-1} \text{CE}\left(\sigma(V(s_t)), \underbrace{\sigma(\hat{V}(s_t))}_{\text{TD target}}\right) + \text{CE}\left(\sigma(V(s_T)), \underbrace{r_T}_{\text{TD target at terminal state}}\right)$

15:     $\mathcal{L}.\texttt{backward}()$
16:     $\mathcal{L} \leftarrow 0$
17: **end for**

---

**Algorithm 3:** $n$-step TD for PRM training

---

**Notation:** $s_t$: the $t$-th reasoning step; $r_t$: reward at $s_t$; $V(s_t)$: state value at $s_t$; $\sigma$: sigmoid; CE: cross-entropy loss; $\texttt{clamp}$: clamps value in [0, 1]; $n$: TD step size
**Input:** Dataset $\mathcal{D}_{\text{PRM}}$ of trajectories with rewards $\{r_t\}_{t=1}^{T}$; process reward model $\text{PRM}_\phi$ with parameters $\phi$; discount factor $\gamma$; step size $n$; $U$: discounted return over a set of steps
1: Initialize total loss: $\mathcal{L}_{\text{total}} \leftarrow 0$
2: **for** each trajectory $\tau = \{(s_1, r_1), \ldots, (s_T, r_T)\}$ in $\mathcal{D}_{\text{PRM}}$ **do**
3:     **for** $t = 1$ to $T$ **do**
4:         $V(s_t) \leftarrow \sigma(\text{PRM}_\phi(s_t))$
5:         $U \leftarrow 0$
6:         **for** $k = 0$ to $n - 1$ **do**
7:             **if** $t + k \leq T$ **then**
8:                 $U \leftarrow U + \gamma^k \cdot r_{t+k}$
9:             **end if**
10:         **end for**
11:         **if** $t + n \leq T$ **then**
12:             $V(s_{t+n}) \leftarrow \sigma(\text{PRM}_\phi(s_{t+n}))$
13:             $U \leftarrow U + \gamma^n \cdot V(s_{t+n})$
14:         **end if**

15:         $\mathcal{L}_t \leftarrow \text{CE}\left(V(s_t), \underbrace{\texttt{clamp}(U, 0, 1)}_{\text{TD target}}\right)$

16:         $\mathcal{L}_{\text{total}} \leftarrow \mathcal{L}_{\text{total}} + \mathcal{L}_t$
17:     **end for**
18:     $\mathcal{L}_{\text{total}}.\texttt{backward}()$
19:     $\mathcal{L}_{\text{total}} \leftarrow 0$
20: **end for**

---

for questions). For MATH-500, we use $\texttt{RLHFlow/Mistral-MATH500-Test}$ [6] from $\texttt{hugging-face}$, which contains 1,024 outputs for each question, generated with a $\texttt{Mistral-7B}$ model [16], which

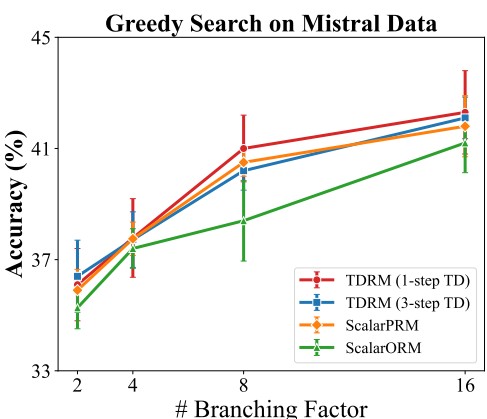

Figure 7: Results of greedy search on our PRM with TD.

has been fine-tuned on `MetaMath` [48]. For Greedy Search, we set the sampling temperature and backbone as $0.4$ and Qwen2.5-Math-7B for generation, and run all the experiments three times to mitigate randomness.

**Dataset for `TDRM` Training.** For TD-based PRM training, we use the `RLHFlow/Mistral-PRM-Data`, which contains step-by-step reasoning trajectories with corresponding correctness labels for intermediate steps. For online RL training, we utilized MATH Level-3 data [11], which comprises 2,500 problem prompts designed to evaluate advanced mathematical reasoning capabilities.

**Baselines.** In verification experiments, we train our baseline using the same Cross-Entropy loss, whereas the target is instead the hard label $Y$. ScalarORM refers to training with only the terminal state, and ScalarPRM incorporates both the intermediate and terminal states. For this setting, we train a ScalarORM using `RLHFlow/Mistral-ORM-Data` and a ScalarPRM using `RLHFlow/Mistral-PRM-Data`. For RL training comparisons, we benchmark `TDRM` with SimpleRL-zoo [50], GRPO in DeepSeek-Math [32], Dr.GRPO [24], and OpenReasoner-Zero [13].

**RL Training Setting.** We conduct online RL training across 4 series of models, including Qwen2.5-(0.5B, 1.5B) [44], GLM4-9B-0414, GLM-Z1-9B-0414 [8], Qwen2.5-Math-(1.5B, 7B) [45], and DS-R1-Distill-Qwen-(1.5B, 7B) [5]. In the `TDRM` framework, the coefficient of $R_{\text{PRM}}$ is set to $a = 0.2$. We run training with a total batch size of 56, divided equally across 7 GPUs, yielding 8 samples per GPU, to optimize compute and efficiency. The rest is used for online sampling with the number of rollouts set to 7, max completion length of 2048, and 1 epoch to mitigate overfitting risks. The RL training framework of `TDRM` is developed from `huggingface/trl`.

In our policy training experiments, aside from the same number of responses for each prompt, 7, to estimate group relative advantage, we allocate different compute resources according to the model size, which leads to a choice of different batch sizes. For 7B and 3B models, we use 8 GPUs, 1 for sampling and 7 for training. The global batch size is 7 (number of devices for gradient calculation) $\times$ 8 (per device batch size) $= 56$. For 1.5B models, we use 4 GPUs, and the global batch size is $3 \times 14 = 42$. For 0.5B models, we use 2 GPUs, and the global batch size is $28 \times 1 \times 2$ (gradient accumulation steps) $= 56$. For `TDRM` training, we use a global batch size of 8 (number of devices) $\times$ 16 (per device batch size) $\times$ 2 (gradient accumulation steps) $= 256$. As for GLM series models, we use `slime`* for training, with a global batch size of 256, and 8 responses for each prompt to estimate group relative advantage.

### F.2 UNDERLYING REWARD DISTRIBUTION

To better understand what a good reward model is like, we visualize the reward distribution over different reasoning steps in Figure 8. This is similar to the smoothness analysis, while it is primarily focused on the distribution of state values, i.e., rewards of `TDRM`. As shown in Figure 8, the trend of reward distribution is similar for both RMs, where it is in a "U" shape as the reasoning step increases,

---

*https://github.com/THUDM/slime

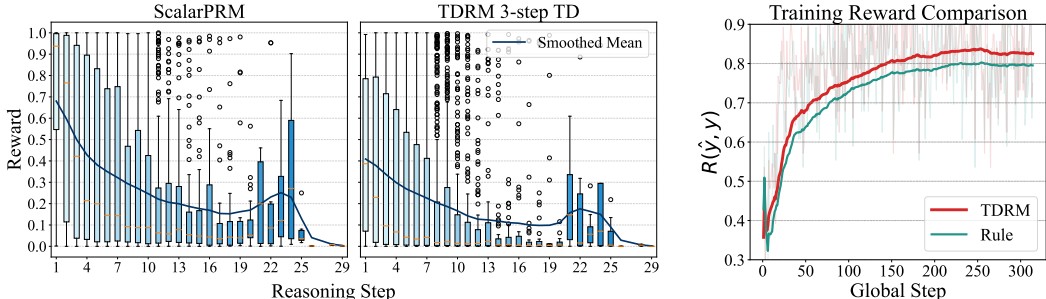

Figure 8: **Left**: Reward distribution over different reasoning steps. TDRM produces more stable and consistent reward estimates, reducing noisy spikes. **Right**: Dynamics of training reward. TDRM consistently yields higher rewards compared to the rule-based baseline, starting from the early steps of training.

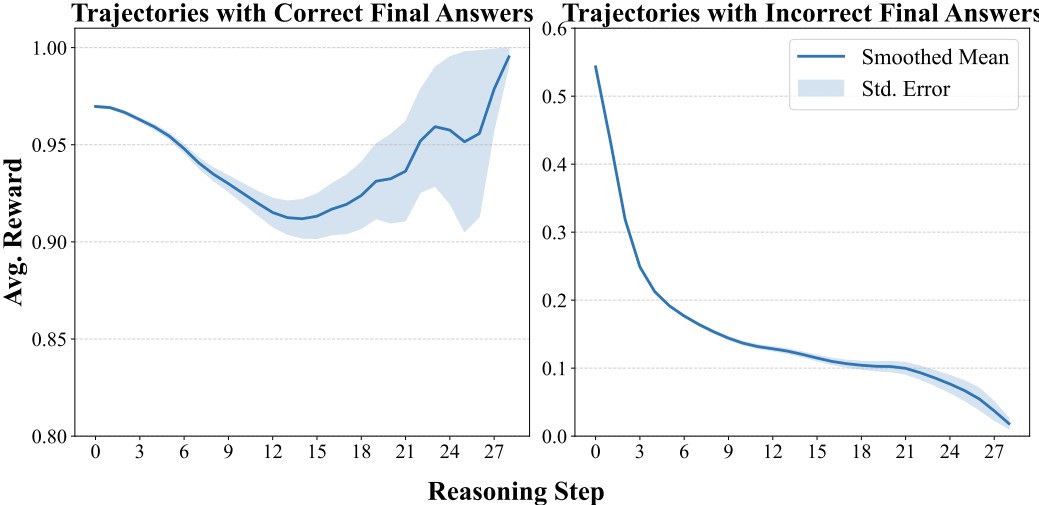

Figure 9: Training reward distribution. This figure shows the smoothed mean reward across different reasoning steps, using trajectories of correct and incorrect final answers separately.

and then drops drastically as the reasoning step becomes much larger. This may reflect the underlying distribution of the dataset that we use to study. However, the distribution of TDRM is smoother and flatter than ScalarPRM, indicating that it is more robust to the number of reasoning steps.

To better understand the underlying mechanism, we first decompose the reward distribution in Figure 8 into two distributions, i.e., a distribution for trajectories with correct final answers and a distribution for trajectories with incorrect final answers. As shown in Figure 9, the reward distribution of trajectories with correct answers exhibits a "U" shape, while that of trajectories with incorrect answers decreases as the number of reasoning steps increases.

### F.3 TEMPLATE USED IN RL TRAINING

> **Prompt for implementation**
>
> <System>
> Please reason step by step, and put your final answer within \boxed{}.
> </System>
> <User>
> Question:
> *Input Question*
> <Assistant>
> Answer:
> Let's think step by step.

### F.4 DETAILS OF VERIFIABLE REWARD

Here we provide the concrete definition of `is_equivalent` and `has_boxed` of $R_{\text{verifiable}}$.

- `is_equivalent`: We only consider boxed answers wrapped within a \boxed{}. And we calculate equivalence after normalizing both $\hat{g}$ and $g$, using a third-party package `mathruler`[†]. If the answers are equivalent then return `True`, otherwise `False`.

- `has_boxed`: To determine if the response has boxed answers and to extract the boxed answers, we use regex ".*\\boxed{.*}.*". If the response matches the regex, then return `True`, otherwise `False`.

### F.5 LOCAL LIPSCHITZ CONSTANT FOR SMOOTHNESS

Inspired by the local Lipschitz Constant, we use the following formula to calculate the smoothness of PRMs:

$$L_{\text{smoothness}} = \frac{1}{|\mathcal{D}|} \sum_{(s_t, s_{t+1}) \in \mathcal{D}} \frac{|V(s_{t+1}) - V(s_t)|}{d(s_t, s_{t+1})}, \tag{12}$$

where $d$ is a function to measure the distance between two adjacent states. Here, we use the cosine similarity of representations from the last hidden state and the last token position. We sample a subset of 1000 trajectories from $\mathcal{D}_{\text{PRM}}$ and compare the constant calculated with state values from TDRM versus ScalarPRM. Empirically, a smaller number of $L_{\text{smoothness}}$ indicates a smoother PRM.

## G STUDY OF TDRM

### G.1 REWARD SMOONTHNESS

To illustrate the comparison of state values obtained from the TDRM and the ScalarPRM, we focus on the difference in their estimates across different quantile bins of $V_{\text{Scalar}}$. The x-axis represents quantile bins of $V_{\text{Scalar}}$, which divides the range of state values computed by the Scalar PRM method into intervals. The y-axis depicts the average difference between the state values derived from TDRM and Scalar PRM, defined as Avg. ($V_{\text{TDRM}}$ - $V_{\text{Scalar}}$). A negative value on the y-axis indicates that TDRM estimates lower state values compared to ScalarPRM in the corresponding quantile bin, while values closer to zero indicate smaller differences between the two methods.

From the Figure 10, it can be observed that:

• Lower Quantile Bins (e.g., (0.0, 0.0075)): The average state value difference is close to zero, meaning that TDRM and ScalarPRM compute nearly identical state values for smaller $V_{\text{Scalar}}$ values.

• Higher Quantile Bins (e.g., (0.412, 0.746) and beyond): The difference becomes significantly negative, indicating that TDRM tends to substantially reduce the state value for states that are assigned larger $V_{\text{Scalar}}$ values by ScalarPRM.

---

[†]https://github.com/hiyouga/MathRuler/tree/main

In conclusion, the ability of the TDRM to reduce the values of high-reward states can be instrumental in achieving smoother rewards during process reward model training or promoting policy robustness in reinforcement learning tasks.

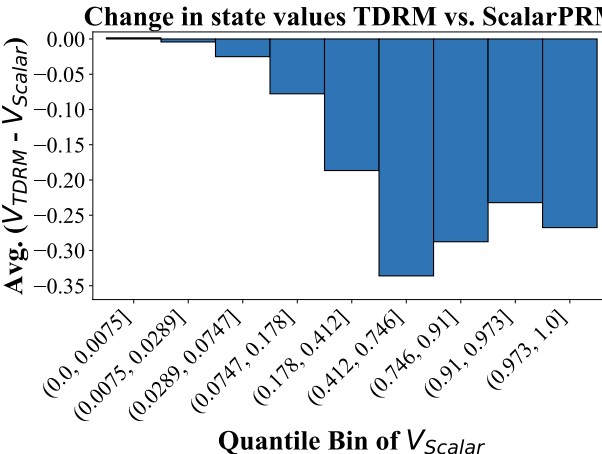

Figure 10: Comparison of state values from the TDRM and the ScalarPRM.

### G.2 ABLATION STUDY OF TRAINING MODELS

**Accuracy and Reward Score in RL.** To further demonstrate the effectiveness of TDRM in RL training, we show a comparison between TDRM and checkpoints trained with pure verifiable reward. We compare the verifiable reward during training for TDRM and the pure rule-based method in Table 9. We can observe that:

• Effectiveness of Rule-based Approaches. Adding rule-based methods consistently improves average performance. For example, in "DS-R1-Distill-Qwen-1.5B", the accuracy increases from 39.2% to 43.2% (+4.0%). This demonstrates the critical role of the rule-based method in mathematical reasoning.

• Impact of PRM trained with TD. The addition of our trained PRM further enhances performance over the purely rule-based approach, especially for larger models. For instance, in "Qwen2.5-Math-1.5B", the average improves from 35.8% (Rule-based) to 36.6% (+0.8%) with trained PRM. This shows PRM's ability to refine value estimation and align intermediate reasoning steps.

• Performance of Ours. Combining rule-based methods and PRM ("Ours") achieves the best results across all settings. Notably, "DS-R1-Distill-Qwen-7B" reaches an average accuracy of 67.4% with a relative improvement of 3.2%, and "Qwen2.5-Math-1.5B" improves by +2.1%, highlighting the synergy of the two components in enhancing smoothness and temporal consistency.

Table 9: Ablation study of general models and reasoning models on the mathematical benchmarks.

| Model | Data Size | MATH 500 | Minerva Math | Olympiad Bench | AIME24 (Pass@1) | AMC23 | Avg. |
|---|---|---|---|---|---|---|---|
| **Backbone is Base Model, Qwen Series** | | | | | | | |
| **Qwen2.5-0.5B** | - | 15.8 | 4.8 | 2.8 | 0.0 | 12.5 | 7.2 |
| + Rule-based | 2.5k | 29.8 | 4.0 | 7.0 | 0.0 | 12.5 | 10.7 |
| + w/ PRM | 2.5k | 7.8 | 1.5 | 1.5 | 0.0 | 7.5 | 3.7 |
| + Ours | 2.5k | 26.2 | 4.8 | 7.1 | 0.0 | 15.0 | **10.8** (**+0.9%**) |
| **Qwen2.5-1.5B** | - | 29.6 | 6.6 | 6.5 | 0.0 | 12.5 | 11.0 |
| + Rule-based | 2.5k | 58.0 | 12.1 | 18.7 | 0.0 | 27.5 | 23.3 |
| + w/ PRM | 2.5k | 16.0 | 4.8 | 5.9 | 0.0 | 15.0 | 8.3 |
| + Ours | 2.5k | 52.8 | 9.9 | 17.8 | 3.3 | 35.0 | **23.8** (**+2.2%**) |
| **Backbone is Chat Model, GLM Series** | | | | | | | |
| **GLM4-9B-0414** | - | 65.8 | 36.8 | 28.7 | 10.0 | 42.5 | 36.8 |
| + Rule-based | 2.5k | 74.0 | 38.6 | 36.3 | 6.7 | 47.5 | 40.6 |
| w/ PRM | 2.5k | 68.2 | 39.3 | 33.5 | 10.0 | 35.0 | 37.2 |
| + Ours | 2.5k | 72.2 | 37.1 | 32.0 | 20.0 | 47.5 | **41.8** (**+3.0%**) |
| **Backbone is Reasoning Model, GLM Series** | | | | | | | |
| **GLM-Z1-9B-0414** | - | 93.6 | 43.8 | 65.5 | 73.3 | 92.5 | 73.7 |
| + Rule-based | 2.5k | 95.6 | 43.4 | 65.2 | 73.3 | 97.5 | 75.0 |
| + w/ PRM | 2.5k | 95.6 | 47.4 | 67.0 | 70.0 | 97.5 | 75.5 |
| + Ours | 2.5k | 94.6 | 44.9 | 66.5 | 80.0 | 97.5 | **76.7** (**+1.6%**) |
| **Backbone is Base Model, Qwen-Math Series** | | | | | | | |
| **Qwen2.5-Math-1.5B** | - | 42.2 | 8.8 | 27.0 | 10.0 | 37.5 | 25.1 |
| + Rule-based | 2.5k | 67.6 | 21.3 | 31.0 | 6.7 | 52.5 | 35.8 |
| +w/ PRM | 2.5k | 63.8 | 19.9 | 26.7 | 16.7 | 50.0 | 35.4 |
| + Ours | 2.5k | 66.2 | 18.4 | 30.1 | 13.3 | 55.0 | **36.6** (**+2.1%**) |
| **Qwen2.5-Math-7B** | - | 63.6 | 12.5 | 25.8 | 13.3 | 42.5 | 31.5 |
| + Our Template | - | 68.8 | 16.2 | 31.1 | 13.3 | 62.5 | 38.4 |
| + Rule-based | 2.5k | 73.2 | 25.0 | 37.8 | 23.3 | 65.0 | 44.9 |
| + w/ PRM | 2.5k | 58.0 | 22.4 | 20.6 | 6.7 | 30.0 | 27.5 |
| + Ours | 2.5k | 74.6 | 26.8 | 37.3 | 36.7 | 62.5 | **47.6** (**+6.0%**) |
| **Backbone is Reasoning Model, DeepSeek Series** | | | | | | | |
| **DS-R1-Distill-Qwen-1.5B** | - | 70.6 | 26.5 | 32.1 | 16.7 | 50.0 | 39.2 |
| + Rule-based | 2.5k | 75.4 | 26.8 | 36.1 | 20.0 | 57.5 | 43.2 |
| + w/ PRM | 2.5k | 69.8 | 18.0 | 30.5 | 33.3 | 45.0 | 39.3 |
| + Ours | 2.5k | 79.8 | 30.5 | 38.2 | 30.0 | 70.0 | **49.7** (**+15.0%**) |
| **DS-R1-Distill-Qwen-7B** | - | 88.0 | 43.0 | 49.9 | 63.3 | 82.5 | 65.3 |
| + Rule-based | 2.5k | 89.6 | 46.0 | 52.4 | 50.0 | 82.5 | 64.1 |
| + w/ PRM | 2.5k | 84.2 | 44.5 | 45.8 | 46.7 | 77.5 | 59.7 |
| + Ours | 2.5k | 91.8 | 50.4 | 54.1 | 53.3 | 87.5 | **67.4** (**+3.2%**) |

# H  CASE STUDY

---

### An Example of Sampled Answer with Reward During RL Training

**Question:** Evaluate $\lfloor\sqrt{12}\rfloor^2$

**Answer from LLM:**

*→ Clear plan provided before calculation.*
To solve the problem, we need to break it down into a sequence of steps:

1. Compute $\sqrt{12}$.
2. Compute the floor value of $\sqrt{12}$, denoted as $\lfloor\sqrt{12}\rfloor$.
3. Square the result from step 2.

*→ Code is used to verify and trace all intermediate values.*
Let's go through each step using Python code to ensure accuracy.

---

#### Python Code

```python
→ Clean and well-commented code for reproducibility.
import math
# Step 1: Compute the square root of 12
sqrt_12 = math.sqrt(12)

# Step 2: Compute the floor value of the square root of 12
floor_sqrt_12 = math.floor(sqrt_12)

# Step 3: Square the floor value
result = floor_sqrt_12 ** 2

print((sqrt_12, floor_sqrt_12, result))
```

*→ Output includes intermediate values, adding transparency.*
(3.4641016151377544, 3, 9)

*→ Step-by-step interpretation of output builds trust in the logic.*
Here is the step-by-step result:

1. $\sqrt{12} \approx 3.464$.
2. $\lfloor\sqrt{12}\rfloor = 3$.
3. $3^2 = 9$.

*→ Final boxed result clearly highlights the answer.*
Thus, the final answer is: $\boxed{9}$

**Reward:** 2.58 *→ High reward reflects well-structured and interpretable reasoning.*

