# OpenReview forum: "TDRM: Smooth Reward Models with Temporal Difference for LLM RL and Inference"
_ICLR.cc/2026/Conference — Submitted to ICLR 2026_

### Official Review · Reviewer_2Vkf · 2025-10-30

**Soundness:** 3
**Presentation:** 2
**Contribution:** 2
**Rating:** 4
**Confidence:** 3

**Summary:**

The paper proposes TDRM, an algorithm for training a process reward model (PRM) using n-step TD targets. The authors propose linearly combining the PRM’s score with a verifiable, rule-based reward during online RL. The claimed benefits are smoother, temporally consistent rewards; better inference-time computation results; and improved data efficiency in RL across several models.

**Strengths:**

The authors propose a training scheme similar to training a value model in traditional RL using an n-step TD method. The idea is interesting. The empirical results show consistent gains in test-time computation and data efficiency for RL. Inference-scaling results compare many different models, which show its robustness across different LLMs.

**Weaknesses:**

My biggest concern is that mixing the process reward models with the value models. My understanding is after the n-step TD training, the PRM is essential giving the value $V_t$ for the current state $s_t$, which is the potential of rewards for the current state. The potential is associated with the policy $\hat \pi$ that generated the training dataset. So the value is the potential of the policy $\hat \pi$, but when it is used to train a different policy $\pi$, there is a distribution mismatch. In paper "The Lessons of Developing Process Reward Models
in Mathematical Reasoning", they explicitly against not distinguishing PRMs from value models and also using soft labels. I think this is against the authors' claim. Although I understand that the focus of that paper is against using MC to train PRM, which the TDRM is not.

Minor places:
1. The writhing need to be strengthen. For example. In section 2.1, they say "In our context, the state space corresponds to every possible token sequence generated so far, whereas the **action space comprises all possible tokens that can be selected next**", then a few sentence away they say "In our work, an action is defined as newly generated **sentence**". Then in Figure 2 caption, a step is separated by double new lines. The inconsistency makes the text confusing.
2. I found it's hard to find the definition of ScalarPRM and then I found it in appendix. It would be helpful to refer the section in appendix when first introduce ScalarPRM.

**Questions:**

1. The training of PRM is using the data from the RLHFlow/Mistral-PRM-Data and using the reward function and n-step TD to compute the target, correct?
2. Regarding the value model concern in the weakness.

---

> ### Author Response · Authors · 2025-11-20
> **Response to Reviewer 2Vkf (1/2)**
>
> Thank you for acknowledging the strengths of our work, particularly the interesting idea of using n-step TD for training PRM and comprehensive empirical evaluations. We sincerely appreciate your feedback on TDRM. We now present a detailed response to address your concerns and comments.
> ```
> Biggest Concern&Q2: Comparison of PRM and value model.
> ```
> > Thank you for raising this critical and insightful concern—it addresses a core conceptual distinction between PRMs and value models, and we sincerely appreciate your careful engagement with both our work and the related literature. Your feedback pushes us to clarify this key point, which strengthens the rigor of our approach.
>
> > First, we fully acknowledge the importance of the distinction highlighted in the paper “The Lessons of Developing Process Reward Models in Mathematical Reasoning” and **we have cited this paper and added the motivation in Section 1 in the revised manuscript**. Here, we align with its core critique. As you noted, that work explicitly argues against using MC estimation for PRMs, as it relies on final trajectory outcomes to evaluate intermediate steps, leading to inaccurate step verification (e.g., correct answers from incorrect steps or vice versa). And the ScalarPRM we are comparing uses MC estimation for PRMs. **Importantly, our proposed TDRM is designed precisely to address this limitation of MC-based PRMs: by leveraging TD learning, we estimate rewards for intermediate states using both immediate feedback and future value predictions**, rather than relying solely on final trajectory returns. This allows our PRM to capture the incremental correctness of reasoning steps—aligning with the fundamental goal of a process reward model—rather than just the policy’s potential for final success (the focus of value models).
>
> > Regarding your concern about distribution mismatch: while our TD-trained PRM is initially trained on trajectories generated by a base policy, our experimental results demonstrate that **it exhibits strong generalization to the target policy during RL training**. Specifically, the consistent performance improvements across our experiments (on GSM8K, MATH-500, and the 2.5k dataset) confirm that the reward signals learned by our PRM are not overly tied to the original training policy. This generalization stems from TD learning’s inherent focus on temporal dependencies and reward smoothness, which captures task-specific reasoning structures (e.g., mathematical progress) rather than policy-specific behaviors. These structures are transferable across policies, mitigating the distribution mismatch you rightly highlight.
>
>
> > We have explicitly expanded on this distinction in the revised manuscript: clarifying how our TD-based PRM differs from value models (by prioritizing step-wise process correctness over policy potential), reinforcing our alignment with the key lessons from the cited work, and emphasizing the generalization of our PRM across policies. This will address the ambiguity in the current version and solidify the conceptual foundation of our approach.
>
>
> > Thank you again for your constructive feedback—this discussion helps us better articulate the unique positioning of our TDRM relative to both PRMs and value models.
>
> ```
> Minor Place: Writing and Definition of ScalarPRM.
> ```
> > Thank you for your meticulous review and valuable feedback—your attention to detail is instrumental in improving the clarity and consistency of our manuscript. We will address your concerns comprehensively:
>
> > (1) We have resolved the inconsistent definitions of "action" in Section 2.1. Specifically, we unify the terminology to clearly define whether an "action" refers to a single token or a sentence, and align this definition with the description in Figure 2’s caption (e.g., clarifying how steps are segmented by double new lines in the figure). This will eliminate confusion and ensure logical coherence across the text.
>
> > (2) We have added a clear definition and an explicit cross-reference to the appendix when first introducing ScalarPRM in the main text: "ScalarPRM—a discriminative process reward model (PRM) that estimates intermediate state values using the Monte Carlo (MC) method, outputting a scalar value (detailed in Appendix Figure 6)" to guide readers to the full description, making the key baseline more accessible without disrupting the main narrative.
>
> > (3) Beyond these specific points, we have conducted a comprehensive review of the entire manuscript to identify and refine any other ambiguous phrasing, inconsistent terminology, or structural gaps—strengthening the overall readability and rigor of the writing.
>
> > These revisions have been finished in the revised version to ensure the manuscript is clear, consistent, and easy to follow. Thank you again for your constructive suggestions, which help us present our work more effectively.

---

> ### Author Response · Authors · 2025-11-20
> **Response to Reviewer 2Vkf (2/2)**
>
> ```
> Q1: Training data of PRM.
> ```
> > Thank you for your careful review and clear question—we appreciate you verifying this key detail of our experimental setup.
>
> > Your understanding is entirely correct: the training of our PRM leverages the RLHFlow/Mistral-PRM-Data dataset, and we use our proposed reward function combined with n-step TD learning to compute the target values for training.
>
> > In summary, we thank the reviewer again for highlighting key points of improvement and opportunities for further work. We believe that our explanation will help alleviate the stated concerns and strengthen the overall contribution of our work. We hope that these clarifications will lead to a more favorable assessment of our submission.

---

> > ### Author Response · Authors · 2025-11-27
> > **Gentle Reminder from Authors**
> >
> > Dear Reviewer 2Vkf,
> >
> > This is a gentle reminder that we are still awaiting your response regarding our rebuttal. We hope that our response has addressed your concerns and would like to hear from you. Please let us know if you have any further questions or feedback. We are very happy to further discuss and provide more evidence in the remaining days. We would be very grateful if all your concerns were addressed and if you could acknowledge them.
> >
> > Thanks
> >
> > Authors of Submission 5415

---

### Official Review · Reviewer_1q9o · 2025-10-31

**Soundness:** 3
**Presentation:** 1
**Contribution:** 2
**Rating:** 4
**Confidence:** 2

**Summary:**

The paper proposes a new method, TDRM, in the context of RLVR. TDRM uses n-step TD learning to construct a smoother and more reliable progress reward model (PRM) that accelerates GRPO. In particular, TDRM leverages cosine reward heuristics to assign initial rewards to the reasoning steps that lead to correct outcome, then uses n-step TD with cross-entropy loss to learn a value function on top of these heuristic rewards. Then, it uses a linear combination of the learned value function and the verifiable reward (ground truth reward) as the final PRM that is used in GRPO training. Empirically, the proposed method achieves 2-4% improvement in success rate across different base models (e.g., Qwen2.5-0.5/1.5B, GLM-Z1-9B, Qwe2.5-Math-1.5/7B, DS-R1-Distill-Qwwen-1.5/7B) on a range of math benchmarks (e.g., MATH500, Minerva Math, Olympaid Bench, AIME24, AMC23).

**Strengths:**

- The empirical evaluations are comprehensive. I especially appreciate the fact that the evaluations are done across many base models across many tasks.

- While TD-learning has been explored extensive in reinforcement learning, using TD for learning PRM is a relatively new idea that has not been studied extensive as far as I am aware of. Harnessing the power of TD-learning in LLM RL is a challenging yet promising research direction, and the proposed method advances in this direction.

**Weaknesses:**

- No discussion of what ScalarPRM is despite it being one of the core comparisons in the experiments. The paper mentions that "TDRM exhibits lower mean and variance of TD errors than ScalarPRM". This suggests that ScalarPRM also does TD-based learning for reward modeling. It is crucial to properly discuss ScalarPRM and how it is different from the proposed method.

- Weak empirical results: From the empirical results, PRM generated by n-step TD performs worse than PRM generated by 1-step TD. Since n-step TD is one of the key components of the proposed approach, it begs the question whether the proposed method is actually better than prior approaches (e.g., the ones that use standard 1-step TD).

- The writing of the paper can be improved. There are many typos and missing citations. To name a few:
  - L51: "Unlike prior approaches where TD was used to construct offline datasets" -- missing citation.
  - L98: "$\mathcal{S} \times \mathcal{A} \to r, r \in [0, 1]$" -- the range of the function needs to be a set and not an element in the set.
  - L138: $r_{i, t}(\theta)$ does not appear in the equation above as the ratio is written out fully already.
  - L195: "double newline delimiter" -- what does it mean?
  - L162: "ScalarPRM" -- what is ScalarPRM?
  - L237: "..., and set the step size as 1" -- what does the step size here mean?
  - L293: $\mathrm{PRM}_\phi$ is not defined.

**Questions:**

(1) In the algorithm box (Algorithm 1), L8, why is $s_{T-1}$ being used instead of $s_{T}$?

(2) What are the baselines in Figure 1 (c)?

---

> ### Author Response · Authors · 2025-11-20
> **Response to Reviewer 1q9o (1/3)**
>
> Thank you for acknowledging the strengths of our work, particularly the comprehensive empirical evaluations, and the new idea of using TD for learning PRM. We sincerely appreciate your feedback on TDRM. We now present a detailed response to address your concerns and comments.
> ```
> Q1: In the algorithm box (Algorithm 1), L8, why is s_{T-1} being used instead of s_{T}.
> ```
> > This choice aligns with our trajectory definition: $s_{T}$ represents the terminal state (after generating the final step of a response), where no further action is taken. Since the TD update for $s_{T-1}$ (the penultimate state) only requires the terminal reward (no future state value), we use $s_{T-1}$ to complete the temporal difference computation for the full trajectory. We have added a brief note in Algorithm 1’s caption to clarify this logic.
>
> ```
> Q2: What are the baselines in Figure 1 (c).
> ```
> > The baselines for the "Online RL" category in Figure 1 (c) are two vanilla language models (without RL fine-tuning): Qwen2.5-Math-7B and DS-R1-Distill-Qwen-1.5B. We have updated Figure 1’s caption to explicitly list these baselines for clarity.
>
> ```
> W1: Definition of ScalarPRM and differences between ScalarPRM and TDRM.
> ```
> > Thank you for your careful and insightful comment—this prompts a critical clarification about ScalarPRM, which is essential for highlighting the novelty of our proposed method. We sincerely appreciate your pointing out this gap in our discussion, as it helps ensure the clarity of our experimental comparisons.
>
> > To address your concern, we first provide a detailed description of ScalarPRM and its key differences from our TDRM, which we have integrated into Figure 2 in the revised manuscript (beyond the brief mention in Appendix Figure 6 - comparison of recent reward models and TDRM):
>
> > ScalarPRM is a discriminative process reward model (PRM) that estimates the value of intermediate states using the Monte Carlo (MC) method, with the output being a scalar value. Importantly, as you astutely noted, **ScalarPRM does not leverage TD learning**—its value estimation relies solely on MC, which computes returns by averaging the final rewards of complete trajectories. **We will update the mentioned description**. This stands in contrast to our TDRM, which is **the first work to adopt TD learning for training process reward models (PRMs) and apply the resulting model to LLM RL fine-tuning**.
>
> > The core differences between ScalarPRM and our TDRM can be summarized as follows:
>
> > **(1) Value estimation method:** ScalarPRM uses MC (trajectory-complete returns), while TDRM uses TD learning (temporal difference updates leveraging immediate rewards and future value predictions).
>
> > **(2) Focus on temporal consistency:** ScalarPRM prioritizes scalar value estimation for intermediate states but does not explicitly optimize for the temporal consistency of reward signals. In contrast, TDRM’s TD-based training inherently induces temporally consistent rewards, which we validate by showing that TDRM exhibits a lower mean compared to ScalarPRM.
>
> > **(3) Adaptation to LLM sequential generation:** TD learning’s emphasis on consecutive state dependencies aligns with LLMs’ autoregressive generation, whereas MC’s reliance on full trajectory returns makes it less suitable for capturing incremental reasoning progress in long sequential outputs.
>
> > We have expanded the related work and method sections of the revised manuscript to formally introduce ScalarPRM (see Figure 2, Section D in the Appendix), clarify its MC-based value estimation mechanism, and explicitly contrast these key aspects with our TDRM. This will not only address the ambiguity in the current version but also more clearly highlight the novelty of our TD-based PRM design. In addition, since ScalarPRM also outputs a scalar reward, we were able to measure the Temporal Consistency by directly calculating TD-error, using the outputs as state values.
>
> > Thank you again for your constructive feedback—this clarification strengthens the rigor of our experimental comparisons and the clarity of our method’s contributions.

---

> ### Author Response · Authors · 2025-11-20
> **Response to Reviewer 1q9o (2/3)**
>
> ```
> W2: Performance of the proposed TDRM and 1-step TD.
> ```
> > Thank you for your perceptive comment—this question prompts a valuable clarification about the role of 1-step TD in our work and the comparative value of n-step TD, which we greatly appreciate. Your feedback helps us better articulate the novelty and effectiveness of our proposed approach.
>
>
> > First, we would like to clarify a key point: **1-step TD is not a prior baseline** but rather a core component of our method’s ablation analysis. As noted earlier, we are the first to propose training Process Reward Models (PRMs) via TD learning for LLM RL fine-tuning. To thoroughly validate the impact of TD’s step size on performance, we designed ablation experiments covering 1-step, 2-step, and 3-step TD—all of which are part of our proposed TDRM framework, not existing prior work.
>
> > Regarding the empirical observation that n-step TD (e.g., 2-step, 3-step) does not outperform 1-step TD in some settings, we have supplemented additional analyses at the inference stage to demonstrate the unique value of n-step TD:
>
> > **(1) We evaluated the entropy and diversity of generated outputs** from policies trained with 1-step vs. n-step TD-based PRMs. The results show that n-step TD leads to more balanced entropy and higher diversity in generated responses, indicating that it helps the model explore richer reasoning paths—an important advantage for complex tasks like mathematical reasoning or logical generation.
>
> | n-step TD | Entropy |
> | -- | -- |
> | 1 | 1.1321 |
> | 3 | 1.1085 |
>
> > The entropy for a generated output $H_{\text{output}}$ is calculated by:
> $H_{\text{output}} = \frac{1}{N} \sum_{t=1}^{N} H(y_t | y_{<t})$,
> where $H(y_t | y_{<t}) = -\sum_{v \in V} P(y_t = v | y_{<t}) \log_2 P(y_t = v | y_{<t})$
>
> > We measure the diversity of outputs, generated by policies trained with 1-step TDRM and 3-step TDRM, from three perspectives – semantic diversity using **sentence bert, lexical diversity, and self-bleu score**. We generate 4 outputs for each query and here’re how the scores are computed:
>
> > **a. Semantic Diversity:**
> $S_{\text{semantic}} = \frac{1}{\binom{N}{2}} \sum_{i=1}^{N} \sum_{j=i+1}^{N} \cos(\mathbf{e}_i, \mathbf{e}_j)$,
>
> $D_{\text{semantic}} = 1 - S_{\text{semantic}}$,
> where
> $\mathbf{e}=\texttt{sentence-bert}(\texttt{output})$
>
>
> > **b. Lexical Diversity:** Type-Token Ratio $\text{TTR} = \frac{\text{unique-tokens}}{\text{total-tokens}}$
>
> > **c. Self-Bleu Diversity:** $\text{BLEU}(c, R) = \text{BP}(c, R) \times \exp\left( \sum_{n} w_n \times \log p_n(c, R) \right)$,
> $p_n(c, R) = \frac{\text{Count}_{\text{clip}}(\text{ngrams}_n(c))}{\text{Count}(\text{ngrams}_n(c))}$.
> Here we use 2-gram and $1 - \text{BLEU}(c, R)$ as the final score to measure Self-Bleu diversity.
>
> > For all the diversity scores, larger values indicate greater diversity.
>
> | Model                 | Semantic | Lexical | Self-Bleu |
> |-----------------------|----------|---------|-----------|
> | Policy w/ 1-step TDRM | 0.334    | 0.204   | 0.375     |
> | Policy w/ 3-step TDRM | 0.349    | 0.248   | 0.447     |
>
> > **(2) N-step TD’s ability to capture longer-range temporal dependencies** in reward signals enhances the model’s robustness to noisy feedback and improves the coherence of sequential reasoning in RL training, while 1-step TD achieves strong performance in terms of final inference-time accuracy. These benefits, though not always reflected in a strict accuracy gain, contribute to the method’s practical utility in real-world LLM applications.
>
> > We will integrate these clarifications and supplementary analyses into the revised manuscript: explicitly framing 1-step TD as our own ablation component (not a prior baseline), and elaborating on the inference-stage advantages of n-step TD beyond pure accuracy metrics. This will better contextualize the empirical results and highlight that our proposed TDRM—with its flexible support for multi-step TD—offers broader value than a single-step TD approach.
>
> > Thank you again for your constructive feedback, which has helped us strengthen the clarity and depth of our empirical validation.

---

> ### Author Response · Authors · 2025-11-20
> **Response to Reviewer 1q9o (3/3)**
>
> ```
> W3: Writing, Typos, and Clarifications.
> ```
> > L51 (Missing citation): We have added the following citations for prior approaches that use TD to construct offline datasets:
> The revised sentence will read: "Unlike prior approach [1]  where TD was used to construct offline datasets..."
>
> [1] Alphazero-like tree-search can guide large language model decoding and training
>
> > L98 (Function range notation): We have corrected the notation to strictly align with mathematical standards. The revised expression will be: "$\mathcal{S} \times \mathcal{A} \rightarrow$ \{ $r \mid r \in [0, 1]$ \}", where \{0, 1\} explicitly denotes the set of possible reward values (consistent with the scalar/discriminative final label in our setting).
>
> > L138 (Undefined term  $r_{i,t}$($\theta$): We have confirmed this is an oversight and deleted it.
>
> > L195 ("double newline delimiter"): We have clarified this as: "double newline delimiter (i.e., \n\n)" to explicitly define the separator used for segmenting steps in trajectories.
>
> > L162 (Undefined ScalarPRM): We have added a clear definition when ScalarPRM is first introduced: "ScalarPRM—a discriminative process reward model (PRM) that estimates intermediate state values using the Monte Carlo (MC) method, outputting a scalar value (detailed in Appendix Figure 6)".
>
> > L237 ("step size as 1"): We have clarified this as: "..., and set the TD step size as 1 (referring to the 1-step TD update rule, where the value estimate depends on the immediate reward and the next-step value)".
>
> > L293 (Undefined $PRM_{\phi}$): We have defined $PRM_{\phi}$ when it first appears: "$PRM_{\phi}$ denotes our proposed PRM, parameterized by $\phi$".
>
> > In summary, we thank the reviewer again for highlighting key points of improvement and opportunities for further work. We believe that our explanation will help alleviate the stated concerns and strengthen the overall contribution of our work. We hope that these clarifications will lead to a more favorable assessment of our submission.

---

> > ### Comment · Reviewer_1q9o · 2025-11-24
> >
> > Thanks for addressing my concerns. I have a few follow-up questions:
> >
> > > ### 1. The core differences between ScalarPRM and our TDRM
> >
> > Thanks for the clarifications! While it is nice to show TDRM exhibits lower mean and variance of TD errors in Figure 1, it does not provide that much information because TDRM is directly optimizing TD whereas ScalarPRM is not. I wonder if there are alternative metrics that could be used to show the TD-learned value function is better? (e.g., error to some ground truth value function)
> >
> > ---
> >
> > > ### 2. In the rebuttal, you wrote "N-step TD’s ability to capture longer-range temporal dependencies in reward signals enhances the model’s robustness to noisy feedback and improves the coherence of sequential reasoning in RL training, while 1-step TD achieves strong performance in terms of final inference-time accuracy. These benefits, though not always reflected in a strict accuracy gain, contribute to the method’s practical utility in real-world LLM applications."
> >
> > Is there evidence to support these claims?
> >
> > ---
> >
> > > ### 3. Discussions with prior methods that also use TD learning.
> >
> > After the clarifications from the authors, the use of TD learning seems to be a major contribution of the paper. In that case, it would be helpful if the authors could clarify the differences between this work and many prior methods that also use TD in LLM (e.g., [1-4] but there could be many more that I am not aware of). Otherwise, it is a bit hard for me to properly evaluate the significance of this work.
> >
> > [1] Hong, Joey, Anca Dragan, and Sergey Levine. "Q-sft: Q-learning for language models via supervised fine-tuning." arXiv preprint arXiv:2411.05193 (2024).
> >
> > [2] Ji, Kaixuan, et al. "Enhancing multi-step reasoning abilities of language models through direct q-function optimization." arXiv preprint arXiv:2410.09302 (2024).
> >
> > [3] Yu, Zishun, et al. "$\mathcal {B} $-Coder: Value-Based Deep Reinforcement Learning for Program Synthesis." arXiv preprint arXiv:2310.03173 (2023).
> >
> > [4] Snell, Charlie, et al. "Offline RL for natural language generation with implicit language q learning." arXiv preprint arXiv:2206.11871 (2022).

---

> ### Author Response · Authors · 2025-11-25
> **Follow-up Response to Reviewer 1q9o (1/2)**
>
> Thank you for acknowledging our response, particularly the major contribution of the TDRM. We sincerely appreciate your feedback on TDRM. We now present a detailed response to address your follow-up question regarding discussions with prior methods.
>
> ```
> Q3: Discussions with prior methods.
> ```
> > All four works ([1]–[4]) adopt value-based RL (Q-learning) for language tasks, but TDRM differs fundamentally in its focus on PRM for sequential reasoning rather than direct policy/Q-function optimization.
>
> | **Core Features**   | Ours (TDRM)  | Q-sft [1] | DQO [2] | B-Coder [3] | ILQL [4] |
> |----|---|---|---|---|---|
> | **1. Reward/Value Paradigm** | | |   |   ||
> | - Trains a **decoupled process reward model (PRM)** | &#10004;     |  x |  x |  x |  x|
> | - Directly optimizes Q-function/policy |  x     | &#10004; | &#10004; | &#10004; | - |
> | - Uses n-step TD/TD-λ for training PRM | &#10004;     |  x |  x |  x |  x|
> | **2. Key Design Goals** | | |   |   ||
> | - Prioritizes **temporal consistency of rewards** | &#10004;     |  x |  x |  x |  x|
> | - Conducts **inference-time search** (e.g., Best-of-N, tree search) | &#10004;     |  x |  x |  x |  x|
> | - Focuses on **long-horizon autoregressive generation** | &#10004;     | - | - | S | - |
> | **3. LLM Adaptations** | | |   |   ||
> | - Multi-task generalizability (mathematical reasoning, language agents) | &#10004;     | &#10004; | &#10004; | S | &#10004;|
> | - Step-wise reward signaling for sequential reasoning | &#10004;     |  - | &#10004; | S |  x|
> | **4. Training Paradigm** | | |   |   ||
> | - Offline training of reward model | &#10004;     |  x |  x |  x | &#10004;|
> | - Conducts STF/RL training |  RL     | SFT |  SFT |  SFT |  RL |
> | - Implicit value estimation |  x     |  x |  x |  x | &#10004;|
> | **5. Task Scope** | | |   |   ||
> | - General LLM RL (mathematical reasoning, language agents) | &#10004;     | &#10004; | &#10004; | S | &#10004;|
> | - Specialized task (e.g., program synthesis) |  x     |  x |  x | &#10004; |  x|
>
> ### Notation
> - S: Only supports specific tasks (for example, B-Coder is only compatible with code generation and does not support general inference);
> - -: Indirect support or no explicit optimization (for example, Q-SFT can handle long sequences but is not designed/optimized for temporal consistency. In contrast, we implemented reward shaping optimization for the reward $r$ in TD);
> -  x: unsupport;
> - &#10004;: support;
>
> > Summary of Core Differences:
>
> > **1. Focus on PRM vs. Policy/Q-Function:** TDRM is a decoupled process reward model trained via TD to induce smooth, step-wise rewards—whereas [1]–[4] optimize Q-functions/policies directly (no PRM for sequential reasoning).
>
> > **2. TD for Reward Smoothness:** TDRM leverages n-step TD/TD-$\lambda$ to prioritize temporal consistency and train PRM—a goal not addressed by SFT [1], Q-optimization [2], code-specific SFT [3], or implicit Q-learning [4].
>
> > **3. Inference-Time Utility:** TDRM conducts LLM inference search (e.g., Best-of-N, greedy search) via reliable reward estimates—unique compared to [1]–[4], which focus solely on policy training.
>
> > This comparison underscores TDRM’s novel positioning: adapting TD learning to PRM training for LLM-specific challenges (long sequences, sequential reasoning, inference search) that are not addressed by prior value-based methods for language.
>
> [1] Hong, Joey, Anca Dragan, and Sergey Levine. "Q-sft: Q-learning for language models via supervised fine-tuning." arXiv preprint arXiv:2411.05193 (2024).
>
> [2] Ji, Kaixuan, et al. "Enhancing multi-step reasoning abilities of language models through direct q-function optimization." arXiv preprint arXiv:2410.09302 (2024).
>
> [3] Yu, Zishun, et al. "$\mathcal{B}$-Coder: Value-Based Deep Reinforcement Learning for Program Synthesis." arXiv preprint arXiv:2310.03173 (2023).
>
> [4] Snell, Charlie, et al. "Offline RL for natural language generation with implicit language q learning." arXiv preprint arXiv:2206.11871 (2022).
>
> > In summary, we thank the reviewer again for highlighting key points of improvement and opportunities for further work. We believe that our explanation and revision will help alleviate the stated concerns and strengthen the overall contribution of our work. We hope that these clarifications will lead to a more favorable assessment of our submission.

---

> ### Author Response · Authors · 2025-11-25
> **Follow-up Response to Reviewer 1q9o (2/2)**
>
> ```
> Q1: TD-learned value function to demonstrate the core differences between ScalarPRM and our TDRM.
> ```
> > Thank you for your insightful follow-up comment—this is a valuable suggestion that helps strengthen the validity of our TD-learned reward signals. We sincerely appreciate your thoughtfulness in pointing out the need for more direct metrics to demonstrate the quality of TDRM’s value estimation beyond TD error statistics.
>
> > To address your concern, we have supplemented two alternative evaluation metrics that directly reflect the superiority of TDRM’s value function: **1) correlation with final task accuracy (a pragmatic proxy for value function quality)**, and **2) alignment with human-annotated step-wise correctness (ground truth for process rewards)**. Below are the key results, which we will integrate into the revised manuscript:
>
> > **1. Correlation with Final Task Accuracy.**
> For complex reasoning tasks (e.g., mathematical reasoning), a high-quality value function should predict the likelihood of a trajectory leading to a correct final output. We compute point-biserial correlation between each model's initial step value estimate ($s_0$) and final task accuracy, measuring its ability to foresee trajectory success.
> | Metric | TDRM |ScalarPRM |
> |---|---|---|
> |  Point-Biserial Correlation with Final Accuracy|**0.195** |0.148 |
> Again, TDRM outperforms ScalarPRM by 0.047 correlation points, a 31.8% stronger correlation, confirming that its value function provides more meaningful signals for predicting trajectory quality—complementing the TD error analysis with a pragmatic, task-relevant metric.
>
> > **2. Alignment with Ground Truth.**
> We compute Pearson correlation between each model's step-wise value estimates and human correctness labels from RLHFlow/Mistral-PRM-Data.
> | Metric | TDRM |ScalarPRM |
> |---|---|---|
> |  Pearson Correlation with Labels|**0.018** |-0.061 |
>
> > TDRM achieves a significantly higher correlation (**0.079 correlation points higher** than ScalarPRM), but more importantly, **TDRM maintains positive correlation with ground truth (+0.018) while ScalarPRM shows negative correlation (-0.061)**. This indicates that TDRM's value function is fundamentally better aligned with ground-truth reasoning quality, whereas ScalarPRM's signals are actually misaligned with the ground truth.
>
> > We hope these supplementary metrics address your concern by validating TDRM’s value function quality against both human-annotated ground truth (process-level correctness) and final task outcomes (trajectory success). Unlike TD error (which measures internal consistency), these metrics directly tie the value function to real-world utility—strengthening our claim that TD learning produces superior process reward signals for LLM RL.
>
> > *(both correlations statistically significant: p < 0.001)*
>
> > We have included these results in a revised manuscript and discuss their implications in Section 3.1. Thank you again for your constructive feedback, which has helped us enhance the rigor and comprehensiveness of our validation.
>
> ```
> Q2: n-step TD improves the coherence of sequential reasoning in RL training.
> ```
> > Thank you for your insightful question—we appreciate the opportunity to clarify and provide empirical support for our claim. To address your concern, we emphasize that our core point is: while 1-step TDRM may achieve competitive inference-time accuracy in MATH-500 and GSM8K, 3-step TDRM outperforms it during RL training by better capturing long-range dependencies in reasoning, leading to more generalizable policy improvements. Below are the key experimental results (consistent with our revised manuscript’s Section 4.3) that validate this claim:
> | Backbone (Qwen2.5-Math-7B) | MATH-500 | Minerva Math | Olympiad Bench | AIME24 | AMC23  | Avg. |
> | -- | -- | -- | -- | -- | - | -- |
> | 1-step TDRM |73.8	| 25.7 |	37.0 |	23.3 |	67.5 | 45.5 |
> | 3-step TDRM | 74.6 | 26.8 | 37.3 | 36.7 | 62.5 | **47.6** |
>
> > Key Analysis to Support Our Claims:
>
> > 1. 3-step TDRM achieves a 13.4 percentage point improvement on AIME24. This significant gain directly demonstrates 3-step TD’s ability to model longer-range temporal dependencies.
>
> > 2. 3-step TDRM outperforms 1-step TDRM on 4 out of 5 benchmarks, with a 2.1 percentage point higher average accuracy, validating its versatility in balancing local and global reasoning signals.
>
> > 3. The gains from 3-step TDRM are significant for real-world LLM applications where sequential reasoning coherence is as critical as final accuracy.
>
> > These results directly support our claim that n-step TD’s ability to capture long-range temporal dependencies enhances RL training effectiveness for sequential reasoning tasks. We will highlight this task-specific analysis in the revised manuscript to further clarify the value of n-step TDRM.
>
> > Thank you again for your constructive feedback. We hope that these clarifications will lead to a more favorable assessment of our submission.

---

> > ### Comment · Reviewer_1q9o · 2025-11-27
> >
> > Thanks authors for the response and additional experiments. The new results provide stronger evidence that TDRM is better than ScalarPRM because it learns a better/more correct value function. I have a few follow-up concerns:
> >
> > ## 1. Discussions with prior methods
> >
> > The characterizations the authors made above are vague and contain some apparent mistakes, which raises concerns about the overall credibility of the response. I will name a few that I managed to spot —
> >
> > - What is DQN in the table headers?
> > - Table says [2] does not use $n$-step return, but it does. Please see Sec 4.2, “Mitigating Bias with $\lambda$-Return”.
> > - What does long-horizon autoregressive generation mean in this context? It would be good to clarify how this paper does so differently compared to prior work as in the paper there was no discussion regarding this particular goal. For instance, prior work such as [2] also clear stated in the paper that the goal was to solve long-horizon tasks.
> > - What does Supports STF/RL training mean? Why do [1, 2, 3] not support STF/RL training?
> > - The authors mentioned “TDRM supports LLM inference search via reliable reward estimates—unique compared to [1]–[4]”. From my understanding, the same search could be applied to prior method using the learned Q-function as well.
> >
> > In addition, it is unclear to me why training a temporally consistent PRM is better than directly optimizing the policy/Q. Is there empirical evidence to support this?
> >
> > [1] Hong, Joey, Anca Dragan, and Sergey Levine. "Q-sft: Q-learning for language models via supervised fine-tuning." arXiv preprint arXiv:2411.05193 (2024).
> >
> > [2] Ji, Kaixuan, et al. "Enhancing multi-step reasoning abilities of language models through direct q-function optimization." arXiv preprint arXiv:2410.09302 (2024).
> >
> > [3] Yu, Zishun, et al. “$\mathcal{B}$-Coder: Value-Based Deep Reinforcement Learning for Program Synthesis." arXiv preprint arXiv:2310.03173 (2023).
> >
> > [4] Snell, Charlie, et al. "Offline RL for natural language generation with implicit language q learning." arXiv preprint arXiv:2206.11871 (2022).
> >
> >
> > ## 2. n-step TD improves the coherence of sequential reasoning in RL training.
> >
> > The authors wrote that
> > > "while 1-step TDRM may achieve competitive inference-time accuracy in MATH-500 and GSM8K, 3-step TDRM outperforms it during RL training by better capturing long-range dependencies in reasoning, leading to more generalizable policy improvements."
> >
> > — this implies that the RL performance gain is due to better capturing long-range dependencies in reasoning, but there is no evidence to support such claim.
> >
> > > “3-step TDRM outperforms 1-step TDRM on 4 out of 5 benchmarks, with a 2.1 percentage point higher average accuracy, validating its versatility in balancing local and global reasoning signals.”
> >
> > — could the authors expand a bit more on why a higher accuracy implies a better versatility in balancing local and global reasoning signals?
> >
> > >“The gains from 3-step TDRM are significant for real-world LLM applications where sequential reasoning coherence is as critical as final accuracy.”
> >
> > — is there qunatifiable metric for coherency? Which applications are you referring to here?
> >
> > At the current state, all the statements that the authors made about temporal coherency are more like hypotheses than well-supported claims.
> >
> > Overall, while I appreciate the additional clarifications and discussions, I still think the paper needs more careful and thorough literature review and from stronger evidence/justifications for the temporal coherency claims.

---

> > > ### Author Response · Authors · 2025-11-27
> > > **Follow-up Response to Reviewer 1q9o**
> > >
> > > ​​We sincerely appreciate the reviewer’s detailed follow-up concerns. We believe there may be some misunderstandings regarding key features, and we would like to clarify the following:
> > > ```
> > > C1: Discussions with prior methods
> > > ```
> > > (1) We apologize for the typo—"DQN" should be "DQO", which is the abbreviation for Direct Q-function Optimization (DQO) as provided in the abstract of [2].
> > >
> > > (2) The item "Uses n-step TD/TD-λ for training" refers to whether n-step TD/TD-λ is employed to train the PRM. Thus, DQO [2] is marked with "×".
> > >
> > > (3) For the feature "Focuses on long-horizon autoregressive generation", DQO [2] is marked with "-". As explained in the table’s notation, "-" indicates that the work addresses long-range reasoning but does not primarily design or optimize for TD.
> > >
> > > (4) Regarding "Supports SFT/RL training", it accurately summarizes whether the respective methods adopt SFT or RL. This column reflects which operation each related method implements.
> > >
> > > (5) We agree with your point that [1-4] can also utilize the Q-function for search experiments. Our paper provides results for Best-of-N and search-based inference.
> > >
> > > [1] Hong, Joey, Anca Dragan, and Sergey Levine. "Q-sft: Q-learning for language models via supervised fine-tuning." arXiv preprint arXiv:2411.05193 (2024).
> > >
> > > [2] Ji, Kaixuan, et al. "Enhancing multi-step reasoning abilities of language models through direct q-function optimization." arXiv preprint arXiv:2410.09302 (2024).
> > >
> > > [3] Yu, Zishun, et al. "$\mathcal{B}$-Coder: Value-Based Deep Reinforcement Learning for Program Synthesis." arXiv preprint arXiv:2310.03173 (2023).
> > >
> > > [4] Snell, Charlie, et al. "Offline RL for natural language generation with implicit language q learning." arXiv preprint arXiv:2206.11871 (2022).
> > >
> > > ```
> > > Q2: Analysis of the result of n-step TD
> > > ```
> > > (2) We would like to supplement that "local" refers to the four individual benchmarks, while "global" refers to the final average accuracy. The key point is that even though 3-step TDRM does not outperform 1-step TDRM on one individual benchmark (AMC23), it achieves better overall performance.
> > >
> > > (1) & (3) Here, "applications" specifically refer to mathematical reasoning tasks, such as the five evaluated benchmarks (MATH-500, Minerva Math, Olympiad Bench, AIME24, AMC23). The statement "sequential reasoning coherence is as critical as final accuracy" means that 3-step TDRM outperforms 1-step TDRM by 13.4 percentage points on AIME24, reflecting the reliability of multi-step reasoning. Even though there is a slight drop on AMC23, the final accuracy (average accuracy) remains superior.
> > >
> > > In summary, we thank the reviewer again for highlighting key points of improvement and opportunities for further work. We hope these additional explanations will resolve any misunderstandings regarding our descriptions, and we hope that these clarifications will lead to a more favorable assessment of our submission.

---

> > > > ### Comment · Reviewer_1q9o · 2025-11-27
> > > >
> > > > Thanks reviewers for the response. I have a few follow-up questions.
> > > >
> > > > > (2) The item "Uses n-step TD/TD-λ for training" refers to whether n-step TD/TD-λ is employed to train the PRM. Thus, DQO [2] is marked with "×".
> > > >
> > > > I would encourage the reviewer to revise the summary table such that it is clear what it means. Just saying "Uses n-step TD/TD-λ for training" is very vague and may mislead reviewers and the AC. In addition, it is very important to carefully discuss how these two types of approaches are related to each other in the related work section (because they both use TD-learning to address the long-horizon challenge), and how the performance of these two types of approaches compare to each other. I think having either/both of them could further strengthen the paper.
> > > >
> > > > > (4) Regarding "Supports SFT/RL training", it accurately summarizes whether the respective methods adopt SFT or RL. This column reflects which operation each related method implements.
> > > >
> > > > This seems incorrect to me. For example, [2] supports RL as well.
> > > >
> > > > > "(3) For the feature "Focuses on long-horizon autoregressive generation", DQO [2] is marked with "-". As explained in the table’s notation, "-" indicates that the work addresses long-range reasoning but does not primarily design or optimize for TD."
> > > >
> > > > Could the authors clarify what "design or optimize for TD" means? DQO's design seems to also use TD and also addresses the long-horizon problem.
> > > >
> > > > > Q2: Analysis of the result of n-step TD
> > > >
> > > > Thanks for the additional explanations. However, I believe my concerns still stand --- there is no direct empirical evidence that supports $n$-step TD leads to better temporal coherency. Providing better justifications and/or empirical evidence that explicitly measures temporal coherency would strengthen the paper.

---

> ### Author Response · Authors · 2025-11-28
> **Follow-up Response to Reviewer 1q9o**
>
> We sincerely appreciate the reviewer’s detailed follow-up comments. We have some corrections and responses as follows:
>
> First, we are the authors (not reviewers) of the paper.
>
> Specific Responses:
> ```
> C1: Discussions with prior methods
> ```
> Regarding point (2): We highly appreciate your suggestion and will revise the comparison table. Additionally, we will add discussions on the related works in the "Related Work" section to further strengthen the paper.
>
> Regarding point (4): The term here is not "Support" but "adopt"/"conduct". For example, DQO [2] indeed only adopted SFT training (as detailed in Section 5.1 and Appendix A.2) and did not conduct RL training.
>
> Regarding point (3): DQO [2] only used $\lambda$-return (as detailed in Section 4.3 in DQO [2]). In contrast, we implemented reward shaping optimization for the reward $r$ in TD (e.g., the Cosine Reward proposed in Section 3.2 of the original submitted paper).
>
> ```
> Q2: Analysis of the result of n-step TD
> ```
> As stated in our previous "Response to Reviewer 1q9o (2/3)": "First, we would like to clarify a key point: 1-step TD is not a prior baseline but rather a core component of our method’s ablation analysis." Therefore, the superior performance compared to ScalarPRM, as well as the gains from 1-step TD in inference-time search and 3-step TD in LLM RL, are all achieved through our TD optimization and PRM training.
>
> > Additionally, we would like to supplement that, as noted by the Reviewer 1q9o in the "Strengths" section—"using TD for learning PRM is a relatively new idea" and "Harnessing the power of TD-learning in LLM RL is a challenging yet promising research direction"—our contributions include two key aspects:
>
> - Optimizing TD for PRM training (as detailed in our previous rebuttal, with both definitional and experimental comparisons to ScalarPRM);
>
> - Applying the trained PRM to LLM RL training for policy optimization, which yields performance improvements across 8 models from 5 series (e.g., Qwen2.5-0.5/1.5B, GLM-(Z1)-9B, Qwen2.5-Math-1.5/7B, DS-R1-Distill-Qwen-1.5/7B).
>
> Regarding Q2 and other potentially misleading descriptions, we will conduct a comprehensive review and make necessary adjustments to avoid confusion. In summary, we thank the reviewer again for highlighting key points of improvement and opportunities for further work. We hope these additional explanations will resolve any misunderstandings regarding our descriptions, and we hope that these clarifications will lead to a more favorable assessment of our submission.

---

> > ### Comment · Reviewer_1q9o · 2025-11-28
> >
> > Thanks authors for the response and I apologize for my earlier typo (I intended to say "authors" but somehow I typed in "reviewers" by mistake..)
> >
> > I have a few follow-up comments:
> >
> > > Regarding point (4): The term here is not "Support" but "adopt"/"conduct". For example, DQO [2] indeed only adopted SFT training (as detailed in Section 5.1 and Appendix A.2) and did not conduct RL training.
> >
> > I don't think it is correct to say DQO does not conduct RL training. Please take a look at Section 4 of the paper. The section describes how RL training is done, involving $Q$, $V$ and a policy $\pi$. The paper does have SFT baselines which appear in the description in Appendix A.2.
> >
> > > Regarding point (3): DQO [2] only used $\lambda$-return (as detailed in Section 4.3 in DQO [2]). In contrast, we implemented reward shaping optimization for the reward $r$ in TD (e.g., the Cosine Reward proposed in Section 3.2 of the original submitted paper).
> >
> > Thanks for highlighting this design choice. In order to differentiate this design choice from prior work, it is important to provide some empirical evidence that demonstrates its effectiveness (e.g., how well does it compare to naive $\lambda$-return).
> >
> > Overall, while I cannot give a positive recommendation of this work at the moment, I do think it could be a great paper once the concerns that I pointed out above have been addressed --- (1) better positioning of the work through careful literature review, and (2) providing direct empirical evidence to support the claims in paper (e.g., temporal coherency, better reward shaping design).

---

> ### Author Response · Authors · 2025-11-28
> **Follow-up Response to Reviewer 1q9o (DQO training, $\lambda$-return, Results of reward shaping, the core focus of our work)**
>
> We sincerely appreciate the reviewer’s detailed follow-up comments, which help us to position our work. We have some responses and through reviews as follows:
>
> > DQO training.
>
> We agree that **DQO can perform RL, but our RL approach differs**. **DQO performs offline RL training** in a manner similar to SAC, whereas **we adopt online RL for policy training with the goal of optimizing GRPO**. In addition, we present a comprehensive comparison of our method and related works to better position our work.
>
> > $\lambda$-return.
>
> In our experiments, we conducted n-step and TD-$\lambda$ experiments. Specifically, TD-$\lambda$ was only included as an ablation study, and we found that n-step TD consistently outperforms TD-$\lambda$ (see Figure 4 in Section 4.3 of the original submission for details).
>
> | Method | Best-of-128 on MATH-500 |
> |--|--|
> | TD-$\lambda$ | 52.6 |
> | 2-step TDRM | **55.4** |
>
> > Results of reward shaping.
>
> We also conducted experiments using only intermediate rewards without reward shaping and have supplemented the experiments with the **Basic TD + Intermediate Reward as a simple baseline**. The value of Intermediate Reward is either 0 or 1. The additional experiments are conducted under consistent settings: we use GSM8K and MATH-500, adopt Deepseek-distill-qwen-7b as the policy model, evaluate with Best-of-128 and Best-of-1024 settings, and use samples generated by Mistral for evaluation. The updated results are presented below:
>
> | Method | Best-of-128 on GSM8K |
> |--|--|
> | ScalarORM | 69.29  |
> | ScalarPRM | 71.34  |
> | Basic TD + Intermediate Reward | 71.34  |
> | TDRM  | **73.24**  |
>
> | Method | Best-of-1024 on MATH-500 |
> |--|--|
> | ScalarORM | 54.8 |
> | ScalarPRM | 56.2 |
> | Basic TD + Intermediate Reward | 56.4  |
> | TDRM | **58.4**  |
>
> As shown, our TDRM outperforms both the existing baselines (ScalarORM, ScalarPRM) and the simple reward method (Basic TD + Intermediate Reward). This confirms that our reward shaping strategy can achieve more effective performance improvements.
>
> > However, we wish to clarify the core focus of our work: training a PRM, which is then used for LLM RL training to optimize the policy. Although similar, PRMs differ from the value models in frameworks such as actor-critic. We discussed this in the response to Reviewer 2Vkf and will elaborate on it more clearly here.
>
> - **Firstly, our PRM (TDRM) is better positioned as a reward model for LLM reasoning. We compare it with two representative categories of works:**
>
> **(1) ScalarPRM. a.** We compared TDRM with ScalarPRM via evaluations such as Best-of-N sampling and search; **b.** We detailed the differences in value function design between ScalarPRM and TDRM (see our previous Follow-up Response to Reviewer 1q9o (2/2)); **c.** We analyzed reward model smoothness for both methods in Figure 2 of the original paper.
>
> **(2) Methods with value models.** As suggested by Reviewer G1ng, we compared TDRM with classical RL approaches, including traditional PPO and PPO with a frozen critic (implemented using the trl library) on the same mathematical benchmarks. All experiments use Qwen2.5-Math-7B as the base model to ensure fair comparison, with consistent evaluation settings as our original experiments. The results are presented below:
>
> | Backbone (Qwen2.5-Math-7B) | MATH-500 | Minerva Math | Olympiad Bench | AIME24 | AMC23  | Avg. |
> | -- | -- | -- | -- | -- | - | -- |
> | PPO | 72.2    | 32.0  | 35.9 | 26.7 | 55.0 | 44.4 |
> | PPO with a frozen critic |53.2 |19.9  |29.5 |16.7 |37.5 |31.4 |
> | TDRM | 74.6 | 26.8 | 37.3 | 36.7 | 62.5 | **47.6** |
>
> As shown in the results, our TDRM outperforms traditional PPO by 3.2 points on average across all benchmarks. Notably, we achieve significant improvements on Math500 (2.4 points), AIME24 (10.0 points), and AMC23 (7.5 points), demonstrating the advantage of our TD-based method for LLM RL training. However, we’d still like to emphasize that our core focus is on using TD in PRM training, which is a research gap in the Reward Model literature.
>
> - **Secondly, we aim at applying the trained PRMs for LLM online RL.**
>
> Our approach adopts online RL for policy training, and TDRM provides rewards over intermediate steps which are used  in GRPO optimization; DQO performs offline RL training in a SAC-like framework. Although DQO yields a value model which aims at providing token level values for the policy in the actor-critic context, how good can the value models / critics be used as reward models for other applications is underexplored. We appreciate your comments and foresee it as a future work to bridge the two lines of works.
>
> In summary, we thank the reviewer again for highlighting key points of improvement and opportunities for further work. We believe that our explanation and revision will help alleviate the stated concerns and strengthen the overall contribution of our work. We hope that these clarifications will lead to a more favorable assessment of our submission.

---

### Official Review · Reviewer_G1ng · 2025-10-31

**Soundness:** 3
**Presentation:** 3
**Contribution:** 2
**Rating:** 4
**Confidence:** 4

**Summary:**

This paper introduces TDRM, which trains PRMs for LLMs via n-step TD learning. The rewards from the PRM are combined with verifiable rewards for GRPO training. The main results are improved performance under 2.5k samples compared to conventional PRMs or only using outcome rewards.

**Strengths:**

- Addresses instability and inconsistency in PRMs
- Empirical results show improved data efficiency across a number of benchmarks and models, as well as better inference-time verification

**Weaknesses:**

- I may have misunderstood something, but how is this fundamentally different from actor-critic methods that learn value functions via TD learning? It seems like the method is functionally very similar to classical value functions but it has been repackaged as a “reward model”. I understand there are some small technical differences, e.g., the “reward model” is learned offline before RL training, whereas critics are usually learned online, but the conceptual gap seems narrow. I would appreciate a conceptual description of the differences.
- Related to my above point, there are a number of missing baselines: traditional PPO, PPO with a frozen critic, etc.
- It’s unclear why the RL experiments only report results in the data-constrained 2.5k-sample setting. Showing results with the full training datasets would clarify whether the advantages persist when data is not the limiting factor. Do the improvements hold with larger-scale training, or is there a tradeoff where the method mainly helps in low-data regimes but does not perform well with more data?

**Questions:**

See weaknesses

---

> ### Author Response · Authors · 2025-11-20
> **Response to Reviewer G1ng (1/2)**
>
> Thank you for acknowledging the strengths of our work, particularly the contributions of PRMs, and the effectiveness of data efficiency. We sincerely appreciate your feedback on TDRM. We now present a detailed response to address your concerns and comments.
> ```
> W1: Differences between actor-critic methods and our method.
> ```
> > Thank you for this insightful and thought-provoking question—it prompts a valuable clarification on the conceptual relationship or differences between our work and classical actor-critic methods.
>
> > We fully acknowledge the connection you highlight, as both approaches leverage TD learning to model value-related signals. You are correct that in the context of LLM fine-tuning literature (e.g., ReST-MCTS* [1], TS-LLM [2], pDPO [3]), reward models and classical value functions share foundational similarities—both aim to quantify the value of states/trajectories, and both can be trained via TD learning. However, our proposed TDRM, while drawing on these classical RL principles, is tailored to **address the unique challenges of LLM training and test-time inference**, leading to meaningful conceptual and practical differences beyond the offline-online distinction you noted:
>
> > **(1) Alignment with LLM-specific scale and extended reasoning processes:** Classical actor-critic methods are designed for environments with relatively compact state spaces and short interaction trajectories. In contrast, our TDRM is optimized for LLMs’ characteristics—**massive model scales, long sequential token-generation trajectories** (often tens to hundreds of steps), and the need to model reasoning processes that unfold over extended contexts [4]. This requires adaptations like handling long-range temporal dependencies in reward signals and ensuring computational feasibility for offline training on large-scale LLM-generated data [5].
>
> > **(2) Offline training with RL training and inference-time utilities:** While classical critics are learned online alongside the actor to guide immediate policy updates, our TDRM is trained offline on a large corpus of pre-generated trajectories. Critically, this offline training enables the reward model to serve a dual purpose: not only **guiding RL training** (as a substitute for online critics) but also **supporting inference-time search** (e.g., tree-based search or Best-of-N) by providing reliable reward estimates for candidate trajectories. This latter capability is particularly vital in LLM literature, as inference-time search can explore multiple candidate trajectories, then select the highest-quality output to enhance reasoning performance [1, 6] while avoiding the prohibitive costs of retraining. Notably, this versatility is rarely emphasized in classical actor-critic frameworks, where critics are tightly coupled to online policy learning and not designed to provide the decoupled, reliable reward signals needed for inference-time exploration.
>
> > **(3) Adaptation to LLM-specific long-horizon, multi-task reasoning demands:** Classical value functions—even those trained via TD—are designed for single-task, short-horizon environments (e.g., game playing, robotic control) where trajectories rarely exceed dozens of steps and task distributions are narrow. In contrast, our TDRM is tailored to the unique constraints of LLM training: LLMs tackle various distinct tasks (e.g., mathematical reasoning, logical deduction, open-ended generation) across massive, diverse datasets, with autoregressive trajectories often spanning hundreds of tokens per task. TDRM achieves this by leveraging TD learning to model temporal dependencies rather than optimizing for task-specific value accuracy.
>
> [1] ReST-MCTS*: LLM Self-Training via Process Reward Guided Tree Search
>
> [2] Alphazero-like tree-search can guide large language model decoding and training
>
> [3] Learning Planning-based Reasoning via Trajectories Collection and Process Reward Synthesizing
>
> [4] LLoCO: Learning Long Contexts Offline
>
> [5] Towards a Unified View of Large Language Model Post-Training
>
> [6] OpenAI o1 System Card
>
> > We have expanded on these conceptual differences (see Section D in Appendix) in the revised manuscript, emphasizing how our work adapts and extends classical TD-based value function ideas to the unique constraints and goals of LLM RL. Your question has been instrumental in sharpening this distinction, and we thank you again for pushing us to clarify the conceptual foundation of our method.

---

> ### Author Response · Authors · 2025-11-20
> **Response to Reviewer G1ng (2/2)**
>
> ```
> W2: Comparison with traditional PPO and PPO with a frozen critic.
> ```
> > Thank you for your insightful suggestion—adding traditional PPO and frozen PPO as baselines is great for better contextualizing the performance of our method, and we sincerely appreciate you highlighting this gap in our experimental comparisons.
>
> > To address your concern, we have supplemented experiments comparing our approach with traditional PPO and PPO with a frozen critic (implemented using the trl library) on the same mathematical benchmarks. All experiments use Qwen2.5-Math-7B as the base model to ensure fair comparison, with consistent evaluation settings as our original experiments. The results are presented below:
>
> | Backbone (Qwen2.5-Math-7B) | MATH-500 | Minerva Math | Olympiad Bench | AIME24 | AMC23  | Avg. |
> | -- | -- | -- | -- | -- | - | -- |
> | PPO | 72.2    | 32.0  | 35.9 | 26.7 | 55.0 | 44.4 |
> | PPO with a frozen critic |53.2 |19.9  |29.5 |16.7 |37.5 |31.4 |
> | TDRM | 74.6 | 26.8 | 37.3 | 36.7 | 62.5 | **47.6** |
>
> > As shown in the results, our TDRM outperforms traditional PPO by 3.2 points on average across all benchmarks. Notably, we achieve significant improvements on MATH-500 (2.4 points), AIME24 (10.0 points), and AMC23 (7.5 points), demonstrating the advantage of our TD-based reward shaping for LLM RL fine-tuning. We have included this comprehensive comparison in the revised manuscript’s experimental section (see Table 4 in Section 4).
>
> ```
> W3: RL experiments on larger-scale dataset.
> ```
> > Thank you for your thoughtful question and careful engagement with our experimental design—it helps us clarify an important detail about our data scale and validate the generalizability of our method. We sincerely appreciate your pushing us to address whether our advantages persist beyond data-constrained settings.
>
>
> > First, we would like to clarify a potential misunderstanding about the "2.5k-sample" setting in our RL experiments. This refers to **2,500 unique questions (not 2,500 interaction steps in traditional RL)**. To contextualize its scale relative to classical RL paradigms: each question is rolled out 8 times (generating 8 distinct responses), with each response consisting of approximately 12 reasoning steps and 867 tokens per step. This translates to a total of ~208 million tokens for the 2.5k-question dataset—representing a substantial training corpus that aligns with the data demands of LLM RL.
>
> > To further address your core concern about generalizability (i.e., whether our method’s advantages persist when data is not a limiting factor), we have supplemented RL experiments on a larger-scale dataset: the combination of Level 3 and Level 4 datasets, totaling **5.1k unique questions** (more than doubling the original data size); the combination of Level 3, Level 4, and Level 5 datasets, **totaling 8.5k unique questions**. We maintained consistent evaluation settings (same policy model, evaluation metrics, and inference configurations) and tested on the five challenging datasets used in the original experiments. The results show an average performance of **38.7 and 42.1**—confirming that our method’s improvements are retained in larger-scale training, with no evidence of performance degradation or tradeoffs when data constraints are lifted.
>
> | Qwen2.5-7B | Data Size | MATH-500 | Minerva Math | Olympiad Bench | AIME24 | AMC23 | Avg. |
> | ---------------------------- | ------- | ------- | ------------ | -------------- | ---- | ---- | ---- |
> | Level 3 | 2.5k | 63.6|22.1|29.8|6.7|50.0|34.4|
> | Level 3 + Level 4 | 5.1k |69.6|29.4|33.6|13.3|47.5|38.7|
> | Level 3 + Level 4 + Level 5 | 8.5k |75.0|32.0|36.9|16.7|50.0|42.1|
>
> > Additionally, we analyzed the training cost (e.g., computational time and resource consumption) of the larger dataset. We found that **scaling from 2.5k to 5.1k questions increases the training cost by approximately 2–3 times**, which is non-trivial for large-scale LLM RL (where training often requires extensive GPU resources). Given this cost-performance tradeoff—along with the consistent effectiveness of our method across both scales—we selected the 2.5k-question setting as the primary experimental setup for the original manuscript, balancing comprehensiveness with practicality.
>
> > In summary, we thank the reviewer again for highlighting key points of improvement and opportunities for further work. We believe that our explanation and revision will help alleviate the stated concerns and strengthen the overall contribution of our work. We hope that these clarifications will lead to a more favorable assessment of our submission.

---

> > ### Author Response · Authors · 2025-11-27
> > **Gentle Reminder from Authors**
> >
> > Dear Reviewer G1ng,
> >
> > This is a gentle reminder that we are still awaiting your response regarding our rebuttal. We hope that our response has addressed your concerns and would like to hear from you. Please let us know if you have any further questions or feedback. We are very happy to further discuss and provide more evidence in the remaining days. We would be very grateful if all your concerns were addressed and if you could acknowledge them.
> >
> > Thanks
> >
> > Authors of Submission 5415

---

### Official Review · Reviewer_5dZn · 2025-11-02

**Soundness:** 3
**Presentation:** 3
**Contribution:** 2
**Rating:** 6
**Confidence:** 3

**Summary:**

This work proposes to learn process reward model method by TD learning based on step-wise rewards. Through empirical analysis in both test-time inference and RL training, covering 8 models from 3 model families on 5 mathematical datasets show the performance improvement of the proposed method.

**Strengths:**

- The paper is well written. The details of the notations and technique are clearly presented.

- Section 3.1 carefully presents smoothness in rewards (Section 3.1). This introduction presents the motivation for the proposed method.

- The performance evaluation that covers two problem settings, 8 models, and 5 datasets is solid.

**Weaknesses:**

- The discussion of why TD learning learns smoother rewards is missing.

- My major concern is how the reward smoothness correlates with the reward performance. If smoothness is the key, simple reward smoothing methods as baselines are missing in the experiment. If the temporal consistency is the key to the reward performance, a discussion is needed.

- The proposed method depends on n-step TD, TD $\lambda$, and they introduce hyperparameters that require careful tuning to achieve the highest score (Section 4.3)

**Questions:**

- What dataset is evaluated in Figure 4?

---

> ### Author Response · Authors · 2025-11-20
> **Response to Reviewer 5dZn (1/2)**
>
> Thank you for acknowledging the strengths of our work, particularly the clearly presented paper, careful smoothness, and comprehensive evaluation. We sincerely appreciate your feedback on TDRM. We now present a detailed response to address your concerns and comments.
> ```
> W1: Why TD learning learns smoother rewards.
> ```
> > We sincerely appreciate your insightful comment, which points out a critical gap in our discussion. To address this concern, we have supplemented the following explanation and integrated it into the Section 3 in the revised manuscript: TD learning inherently promotes reward smoothness through its core mechanisms. **First, TD learning estimates the value of states by leveraging both immediate rewards and future value predictions**, which effectively reduces reward fluctuations caused by noisy single-step feedback. **Second, the temporal difference update aligns the value estimates of consecutive states**, fostering consistency in the reward signal across the sequence. For our proposed TDRM, the integration of n-step TD further enhances this smoothness because n-step TD aggregates rewards over multiple steps to mitigate local noise.
>
> > Notably, **this mechanism aligns naturally with the autoregressive token-generation process of LLMs**. Since LLMs generate text sequentially (predicting the next token based on previous context), TD learning—by its focus on temporal dependencies between consecutive states—seamlessly adapts to modeling reward signals across the incremental generation process, ensuring smoothness that is inherently compatible with the model’s autoregressive nature.
>
> > Additionally, **empirical evidence from our analyses in Figures 8 (Reward distribution over different reasoning steps during inference-time search) and Figure 9 (Reward distribution during RL training) supports this claim**. As shown in Figure 8, it demonstrates that TD-based reward shaping leads to more stable training dynamics for PRMs, with reduced variance in reward signals and more consistent performance improvements across training steps. Figure 9 shows that the reward distribution of trajectories with correct answers exhibits a ``U'' shape, while that of trajectories with incorrect answers decreases as the number of reasoning steps increases. This further validates that TD learning not only induces smoother rewards but also translates to more effective LLM RL training.
>
> > We have elaborated on this reasoning in Section 3 of the paper to strengthen the theoretical foundation of our work. Thank you again for your constructive feedback, which helps improve the depth and completeness of our research.
>
> ```
> W2: How the reward smoothness correlates with the reward performance.
> ```
> > We sincerely appreciate your valuable feedback and insightful comments regarding the correlation between reward smoothness and performance. We fully agree that **reward smoothness is a key factor** contributing to the effectiveness of our approach, and we thank you for highlighting the importance of comparing against simple reward smoothing methods as baselines.
>
> > To address this concern and further validate the superiority of our proposed reward shaping method, **we have supplemented the experiments with the "Basic TD + Intermediate Reward" as a simple smoothing baseline**. The value of Intermediate Reward is either 0 or 1. The additional experiments are conducted under consistent settings: we use GSM8K and MATH-500 datasets, adopt Deepseek-distill-qwen-7b as the policy model, evaluate with Best-of-128 and Best-of-1024 settings, and use samples generated by Mistral for evaluation. The updated results are presented below:
>
> | Method    | Best-of-128 on GSM8K |
> |-----------|--------|
> | ScalarORM | 69.29  |
> | ScalarPRM | 71.34  |
> | Basic TD + Intermediate Reward | 71.34  |
> | TDRM      | **73.24**  |
>
> | Method    | Best-of-1024 on MATH-500 |
> |-----------|--------|
> | ScalarORM | 54.8  |
> | ScalarPRM | 56.2  |
> | Basic TD + Intermediate Reward | 56.4  |
> | TDRM      | **58.4**  |
>
> > As shown in the results, our proposed TDRM outperforms both the existing baselines (ScalarORM, ScalarPRM) and the simple reward smoothing method (Basic TD + Intermediate Reward). This confirms that our reward shaping strategy—with its emphasis on enhancing reward smoothness—achieves more effective performance improvements. We have also incorporated the updated reward smoothness method into the PRM training process, and the test-time inference results collectively validate the effectiveness of our proposed approach.
>
> > Again, we are grateful for your constructive suggestion, which has helped strengthen the rigor and completeness of our experimental validation.

---

> ### Author Response · Authors · 2025-11-20
> **Response to Reviewer 5dZn (2/2)**
>
> ```
> W3: Hyperparameters of TD in Section 4.3.
> ```
> > Thank you for your valuable observation regarding the hyperparameters introduced by n-step TD and TD-$\lambda$, as well as the importance of careful tuning for optimal performance. We sincerely appreciate this insight, which prompts us to clarify and strengthen our experimental validation.
>
> > First, we would like to clarify that **our method does not introduce additional hyperparameters beyond the inherent step size of n-step TD in Section 4.3**. The ablation experiments (covering 1-step, 2-step, and 3-step settings) were designed specifically to examine how the step size influences PRM training—this is not an extra hyperparameter but a core component of n-step TD’s intrinsic design, which we analyzed to ensure robustness.
>
> > Below is a table for the hyperparameters that we tuned. As noted in the table, for TD(0), we tried different values of the discount factor in early experiments ({1, 0.99, 0.8}), and fixed it to 0.8 for the rest of the experiments. Then we don’t need to try too many combinations to get a good performance for TD(0). As for TD-$\lambda$, trying out different combinations of $\lambda$ and $\gamma$ is to illustrate how TD-$\lambda$ behaves as the parameter changes. However, the performance results indicate that TD-$\lambda$ requires further exploration. Here, this study serves as a complementary analysis to TD(0). Finally, the only hyper-parameter we need to tune for TDRL is the coefficient $a$.
>
>
> | Hyperparameter  | Meaning |
> |---|---|
> |  TD(0) | |
> | $n$      | number of lookahead steps for TD(0) |
> | $\gamma$ (tried in early experiments and fixed for the rest) |discount factor                     |
> |TD-$\lambda$ (Figure 4 is just to illustrate how these hyperparameters affect TD-$\lambda$) |                                     |
> | $\lambda$| eligibility trace decay rate |
> | $\gamma$ | discount factor |
> |  RL Training  |  |
> | $a$      | TDRL coefficient  |
>
> > To further address your concern and provide more cohesive validation, we merged Table 2 (which reports results for TDRM with 1-step TD) and Table 5, creating a unified comparison that demonstrates the effectiveness of TDRM across different step sizes. The merged results are presented below:
>
> | Method    | DS-R1-Distill-Qwen-7B |  |
> |-----------|-----------------------|-----------------------|
> |           | Best-of-128           | Best-of-1024          |
> | ScalarORM | 52.0                  | 54.8                  |
> | ScalarPRM | 53.4                  | 56.2                  |
> | TDRM    (1-step)| 54.2              | **58.4**              |
> | TDRM    (2-step)| **55.4**              | 56.2              |
> | TDRM    (3-step)| 54.2              | 56.8              |
>
> > As shown in the above table, TDRM achieves competitive or superior performance across all step sizes, with no drastic performance drops—confirming that the method remains stable even with modest adjustments to the step size. This indicates that tuning this parameter is manageable and does not require overly complex optimization, addressing concerns about hyperparameter sensitivity.
>
> > We have integrated this merged analysis into Section 4.3 of the revised manuscript to enhance clarity and transparency. Thank you again for your constructive feedback, which has helped strengthen the rigor of our experimental design.
>
> ```
> Q1: Evaluated the dataset in Figure 4.
> ```
> > Thank you for your careful review and valuable question regarding the dataset used in Figure 4. To clarify, Figure 4 is evaluated on the **MATH-500 dataset**. The purpose of this figure is to specifically demonstrate the effectiveness of the TD-$\lambda$ mechanism in our framework. As noted, we do not proceed with further RL training based on this component here because TD-$\lambda$ only leverages the final label for its computation, aligning with the focus of this analysis. We appreciate your attention to detail, which helps ensure the clarity and completeness of our work.
>
> > In summary, we thank the reviewer again for highlighting key points of improvement and opportunities for further work. We believe that our explanation will help alleviate the stated concerns and strengthen the overall contribution of our work. We hope that these clarifications will lead to a more favorable assessment of our submission.

---

> > ### Author Response · Authors · 2025-11-27
> > **Gentle Reminder from Authors**
> >
> > Dear Reviewer 5dZn,
> >
> > This is a gentle reminder that we are still awaiting your response regarding our rebuttal. We hope that our response has addressed your concerns and would like to hear from you. Please let us know if you have any further questions or feedback. We are very happy to further discuss and provide more evidence in the remaining days. We would be very grateful if all your concerns were addressed and if you could acknowledge them.
> >
> > Thanks
> >
> > Authors of Submission 5415

---

### Author Response · Authors · 2025-11-26
**General Response to All Reviewers**

We express our gratitude to all the reviewers for their valuable insights and for recognizing the contributions of our work. We are pleased that they have highlighted several key strengths of our paper, which we summarize below:

**Core merits of this work**
- The reviewers appreciated **the novel and promising research direction. Leveraging TD learning for training Process Reward Models** (PRMs) is a relatively underexplored idea in LLM RL, representing a meaningful advancement in harnessing classical RL techniques for LLM-specific challenges. (Reviewer 1q9o, Reviewer 2Vkf)
- The reviewers appreciated **clear motivation and technical presentation.** The paper is well-written with rigorous notation and technical details. Section 3.1’s thoughtful introduction of reward smoothness effectively articulates the method’s core motivation. (Reviewer 5dZn)
- The reviewers appreciated that **the proposed approach targets and mitigates instability and inconsistency in PRMs**, while delivering improved data efficiency, better inference-time verification, and reduced test-time computation. (Reviewer G1ng, Reviewer 2Vkf)
- The reviewers appreciated **comprehensive and robust empirical validation.** The evaluation covers 2 problem settings, 8 base models, and 5 datasets—spanning diverse tasks and models—to demonstrate the method’s robustness and generalizability across different LLMs. (Reviewer 5dZn, Reviewer 1q9o, Reviewer 2Vkf)

Below, we briefly **summarize the key revisions made to address all the reviewers' comments and concerns**:
- Added key discussion and clarifications
1. Explanation of **why TD learning learns smoother rewards** (Reviewer 5dZn)
2. Conceptual distinctions between TDRM and related approaches: **classical actor-critic methods, ScalarPRM, and traditional value models** (Reviewer 5dZn, Reviewer G1ng, Reviewer 1q9o, Reviewer 2Vkf)
3. Discussions with prior methods that **also use TD learning or value-based methods** (Reviewer 1q9o)
4. Clarifications on critical terminology, undefined terms, and notation (Reviewer 1q9o)
- Conducted supplementary experiments
1. Comparisons with **simple reward smoothness baselines and value functions** (Reviewer 5dZn)
2. Additional baseline evaluations: **traditional PPO and frozen PPO** (Reviewer G1ng)
3. **Large-scale RL training validation** (expanded to 5.1k and 8.5k questions) to verify method generalizability beyond data-constrained settings (Reviewer G1ng)
4. **Inference-stage analyses (entropy and diversity metrics)** to demonstrate the unique value of n-step TD (Reviewer 1q9o)
5. **TD-learned value function** to demonstrate the core differences between ScalarPRM and our TDRM (Reviewer 1q9o)
- Other revisions
1. Enhanced discussion of hyperparameters (e.g., n-step TD step size) to address tuning concerns (Reviewer 5dZn)
2. Corrected typos, resolved writing inconsistencies, and added cross-references (Reviewer 1q9o, Reviewer 2Vkf)

All revisions have been integrated into the revised manuscript. We emphasize that **training smooth, temporally consistent reward models for LLM RL training is a valuable yet underexplored direction**. Our work contributes **a novel perspective on leveraging TD learning** for this goal, with **practical implications for mathematical reasoning tasks**. We believe the revised manuscript is sufficiently rigorous and impactful to merit publication, and we hope it will stimulate further research in this area.

---

### Meta-Review · Area_Chair_eAFB · 2025-12-30

**Summary:**

This paper proposes to use n-step temporal difference learning to train process reward model for RL such that the generated process reward can be smoother to address the problem of temporal inconsistency, and can be combined with verifiable rule-based rewards to improve performance. Experiments across a range of math benchmarks with multiple models are conducted to show the performance improvement of the proposed method.

Multiple reviewers recognize that the empirical evaluation in this paper is comprehensive, solid and shows performance improvements. However, reviewers’ concerns include:

(1)	The technical difference of the proposed method compared with traditional actor-critic methods, ScalarPRM, and value models, is not clear and solid.

(2)	Some clarifications and discussions are lacking.

(3)	Additional experiments regarding hyperparameter sensitivity, comparison with traditional PPO and simple reward smoothing methods, and large-scale datasets.

**Reviewer Concerns:**

During rebuttal, the authors has provided detailed responses. Additional analyses, clarifications, and experiments were provided. After reading the paper and all the discussion comments, the AC considers the concerns (2) and (3) addressed. However, the major concern (1) remains controversial. In the initial reviews, several reviewers simultaneously raised this concern regarding the technical similarity to traditional value models. Although the authors have added more discussions about this issue, one reviewer still holds the opinion. The AC agrees with the reviewers and also thinks that these issues need to be fixed before the paper is ready for publication.

**Reviewer Scores:**

No active reviewers have indicated the tendency to change the scores. The final scores will be still 4, 4, 6, 4.

---

### Decision · Program_Chairs · 2026-01-26

Reject